


# Gradients of Column $CO_2$ across North America from the NOAA Global Greenhouse Gas Reference Network

Xin Lan[1,2], Pieter Tans[1], Colm Sweeney[1,2], Arlyn Andrews[1], Andrew Jacobson[1,2], Molly Crotwell[1,2], Edward Dlugokencky[1], Jonathan Kofler[1,2], Patricia Lang[1], Kirk Thoning[1], Sonja Wolter[1,2]

[1]National Oceanic and Atmospheric Administration, Earth System Research Laboratory, Boulder, 80303, Colorado, USA

[2] University of Colorado, Cooperative Institute for Research in Environmental Sciences, Boulder, 80309, Colorado, USA

*Correspondence to*: Xin.Lan (xin.lan@noaa.gov)

**Abstract.** This study analyzes seasonal and spatial patterns of column carbon dioxide ($CO_2$) over North America calculated from aircraft and tall tower measurements from the NOAA Global Greenhouse Gas Reference Network from 2004 to 2014. Consistent with expectations, gradients between the eight regions studied are larger below 2 km than above 5 km. The 11-year mean $CO_2$ dry mole fraction ($XCO_2$) in the column below ~330 hPa (~ 8 km above sea level) from NOAA's $CO_2$ data assimilation model, CarbonTracker (CT2015), demonstrates good agreement with those calculated from calibrated measurements on aircraft and towers. Total column $XCO_2$ was attained by combining modeled $CO_2$ above 330 hPa from CT2015 with the measurements. We find large spatial gradients of total column $XCO_2$ during June to August, and the north and northeast regions have ~3 ppm stronger summer drawdown than the south and southwest regions. The spatial gradients of total column $XCO_2$ across North America mainly reflect large-scale circulation patterns rather than regional surface sources and sinks. We have conducted a CarbonTracker experiment to investigate the impact of Eurasian long-range transport. The result suggests that the large summer time Eurasian boreal flux contributes about half of the north-south column $XCO_2$ gradient across North America. Our results confirm that continental-scale total column $XCO_2$ gradients simulated by CarbonTracker are realistic and can be used to evaluate the credibility of spatial patterns from satellite retrievals, such as the long term average spatial patterns from satellite retrievals reported for Europe which show larger spatial difference (~ 6 ppm) and scattered hot spots.

## 1 Introduction

Atmospheric measurements of carbon dioxide ($CO_2$) from ground and airborne platforms have greatly increased our knowledge of the global carbon cycle. Observations of $CO_2$, including the NOAA Global Greenhouse Gas Reference Network (GGGRN), initially emphasized ground-based measurements. These observations, started by C.D. Keeling, have monitored the $CO_2$ trend on both regional and global scales for over 50 years (e.g., Keeling and Rakestraw, 1960; Tans et al., 1989). In addition, the frequency and spatial distribution of airborne measurements have increased rapidly in the last two decades, providing important information about horizontal and vertical variability of atmospheric $CO_2$ (e.g., Gerbig et al., 2003; Choi et al., 2008; Biraud et al., 2013). Routine aircraft



measurements from the NOAA/ESRL GGGRN monitor the large-scale distributions of a suite of trace gases,
including $CO_2$, under the influence of continental processes (Sweeney et al., 2015). A very successful approach has
been to employ commercial aircraft as a platform for $CO_2$ measurements, such as Japan's CONTRAIL
(Comprehensive Observation Network for TRace gases by AIrLiner) project which has provided valuable
information for $CO_2$ in the high troposphere and lower stratosphere (Machida et al., 2002; Machida et al., 2008).
Vertical profiles of atmospheric $CO_2$ reflect the combined influences of surface fluxes and atmospheric mixing.
Vertical profiles are particularly useful for evaluating vertical mixing in atmospheric transport models that are used
for inverse modeling (e.g. Stephens et al., 2007) to derive estimates of regional- to continental-scale $CO_2$ sources
and sinks(e.g., Tans et al., 1990; Gurney et al., 2002; Gurney et al., 2004; Ciais et al., 2010;).
While $CO_2$ sources and sinks are better constrained at the global scale by global mass balance, it remains
challenging to accurately resolve $CO_2$ sources and sinks at regional-to continental-scale, the apportionment of which
depends on relatively minor variations of the observed spatial and temporal patterns of $CO_2$. When averaging over a
few months and longer the largest portion of the variations over continents results from hemispheric-scale terrestrial
uptake (photosynthesis)/emissions (respiration) and fossil fuel emissions, while regional net fluxes can make a
relatively small contribution to the signal. For example, a simple mass balance argument shows that all U.S. $CO_2$
emissions from fossil fuel burning (~1.4 Pg yr$^{-1}$) create a total column enhancement of only 0.6 ppm on average in
air parcels over the East Coast compared to the West Coast and Gulf Coast if we assume a residence time of the
emissions of 5 days to pass the contiguous U.S. (~8×10$^{12}$ m$^2$).
With careful calibration, air handling, and analysis, the uncertainties of in-situ measurements are less than 0.1
ppm. However, in-situ observation networks are sparse in global and regional coverage. Remote sensing data
radically increase the number of observations and capture under-sampled regions. It is likely to have a valuable
impact on our understanding of the carbon cycle. However, both the precision and the potential of even very small
systematic biases in remote sensing measurements need to be carefully evaluated. Vertical profiles from in-situ $CO_2$
measurements have been used to evaluate ground-based total column $XCO_2$ (X stands for dry mole fraction)
determinations, such as those from the Total Carbon Column Observing Network (TCCON) (Washenfelder et al.,
2006; Wunch et al., 2010; Messerschmidt et al., 2011; Tanaka et al., 2012). The uncertainty of TCCON total column
$CO_2$ is reported to be 0.4 ppm (1σ) after comparison to aircraft measurements (Wunch et al., 2010). Vertical profiles
are also used to evaluate satellite retrievals of total column $XCO_2$, such as those from the Tropospheric Emission
Spectrometer (TES)(Kulawik et al., 2013) and the Greenhouse Gases Observing SATellite (GOSAT) (Inoue et al.,
2013, 2016; Saitoh et al., 2016). Satellite retrieval products have known and unknown biases (due to errors in
spectroscopy, viewing geometry, spatial differences in clouds and aerosols, surface albedo, etc.) that can result in
false horizontal gradients in total column $XCO_2$ for inverse estimates of sources (Miller et al., 2007; Crisp et al.,
2012; Feng et al., 2016). After correction for known biases, the GOSAT total column $CO_2$ retrievals biases range
between -2.09 to 3.37 ppm (mean = 0.11 ppm) across different aircraft sites over land, compared with aircraft-based
total column $XCO_2$ (Inoue et al.,2016). By comparing with TCCON, the Orbiting Carbon Observatory-2 (OCO-2)
retrieval of total column $XCO_2$ was estimated to have a mean difference less than 0.5 ppm with RMS differences
typically below 1.5 ppm after bias correction (Wunch et al., 2016). The overall uncertainty of satellite retrievals is



relatively large compared with the total column $XCO_2$ calculated from in-situ measurements. Total column $XCO_2$
calculated from vertical profiles from the Japanese CONTRAIL project (Machida et al., 2008) and from the NOAA
Carbon Cycle and Greenhouse Gas aircraft program (Sweeney et al., 2015) complemented with simulated profiles
from a chemistry–transport model above the maximum altitude of the data have uncertainty less than 1 ppm
(Miyamoto et al., 2013). The relatively small uncertainty of the in situ-based total column XCO2 suggests that they
can be used to evaluate satellite retrievals of column averaged $CO_2$. Since aircraft profiles co-located with satellite
retrievals are rare, it is useful to consider the statistics of total column $XCO_2$ fields derived from repeated aircraft
profiles over particular locations.

The effect of satellite column averaging kernels and a priori profiles when comparing aircraft-based column

$XCO_2$ with GOSAT retrievals has been assessed by Inoue et al. (2013). For the case considered, application of the
averaging kernel and a priori profile to simulate total column $XCO_2$ was generally within ± 0.1 ppm of the density
weighted total column, suggesting that the averaging kernels can only account for small part of the overall
uncertainty of the GOSAT total column $XCO_2$ (Inoue et al., 2013).

Transparent and objective estimates of $CO_2$ sources and sinks derived from atmospheric measurements are

paramount for validating emissions reduction efforts and other mitigation policies, and for lowering the uncertainties
of carbon cycle-climate feedbacks. The latter are major ambiguities in predicting future climate, such as potential
uncontrolled $CH_4$ and $CO_2$ emissions from warming permafrost in Arctic regions. Satellite retrievals of total column
$XCO_2$ can significantly improve estimates of source and sinks only if they are sufficiently precise and accurate
(Rayner and O'Brien, 2001; Houweling et al., 2004), meaning that even very small systematic errors (biases) must
be eliminated. Here, we analyze the spatial and temporal variability of column $CO_2$ over North America using well-
calibrated $CO_2$ measurements from aircraft and tall tower, and we use model results from NOAA's CarbonTracker,
version CT2015 (Peters et al. 2007, with updates documented at http://carbontracker.noaa.gov) to investigate the
primary drivers of variability in total column $XCO_2$. The aircraft data enable direct analysis of column $CO_2$
characteristics, which is the fundamental step for accurate apportionment of sources and sinks. This study focuses on
the long-term averaged column $CO_2$ gradient and the contributions of different vertical layers to the total column
variability. It can serve as a reference for evaluating current and future column $CO_2$ retrievals from both ground and
satellite platforms.
**2 Methods**
**2.1 Aircraft and tall tower sampling**
Aircraft sampling in the NOAA GGGRN intends to provide vertical profiles of long-lived trace gases to capture
their seasonal and interannual variability. The aircraft sampling system consists of 12 borosilicate glass flasks in
each programmable flask package (PFP), a stainless-steel gas manifold system, and a data logging and control.
These flasks (0.7 L each) are pressurized to obtain 2.2 L of sample air from each target altitude. Air samples are then
shipped back to NOAA/ESRL for carefully calibrated and quality-controlled measurements. Carbon dioxide is
measured using a nondispersive infrared analyzer. Long-term measurements at ~15 sites are carried out using light





aircraft that can reach 8.5 km. Air samples are collected mostly during late morning to early afternoon, when the air
mass within the planetary boundary layer (PBL) is generally well mixed, and $CO_2$ enhancement near the ground
from plant respiration during the night has been mixed throughout the boundary layer. Normally, the aircraft follows
a pre-decided route such that most samples are collected within 0.1° of the site location. The sampling frequency
varies from site to site, currently from twice a month to once every 1.5 months.  For more sampling details, quality
control discussions, and an evaluation of the sampling frequency, please refer to Sweeney et al. (2015). More
information on the aircraft sites can be found at http://www.esrl.noaa.gov/gmd/ccgg/aircraft/. We estimate the
uncertainty of individual measurements of $CO_2$ in flask air (68% confidence level) at 0.08 ppm. However, we have
seen evidence of positive biases for samples collected using older flasks that may contain contaminants.   Andrews
et al. (2014) reported biases that increased from <0.1 ppm in 2008 to an average offset in 2013 of 0.36 ppm. The
aircraft sampling protocol was modified starting in August 2014 to mitigate this bias. For samples collected prior the
protocol change, laboratory tests showed that new/clean flasks have zero bias, but some older/dirty flasks could have
biases of > 1 ppm. This bias is not consistent among individual flasks and increasing over time (Andrews et al.,
2014), the potential bias is hard to quantify for measurements before August 2014. Thus, the high bias is not
corrected in our study. More recently, low bias has been found in PFP measurements when the ambient humidity is
high, based on comparisons of PFP measurements with data from in-situ analyzers at tall towers. We are working to
understand and quantify this bias, and for this study we have derived a preliminary correction factor, which shows a
linear trend  with $-1.4$ ppm $CO_2$ offset per 1% above 1.7% of ambient water content (in mole fractions). Only ~ 4%
of total aircraft measurements or ~ 12% of those below 2 km are impacted by humidity higher than 1.7%, for which
we have applied corrections before data analysis. The mean correction applied is $0.53 \pm 0.4$ (1 σ) ppm for the
impacted data.

The NOAA tall tower network measures $CO_2$ and other trace gases within the continental boundary layer.

Continuous in-situ measurements are conducted using nondispersive infrared (NDIR) absorption sensors and cavity
ring-down analyzers. The long-term stability of these systems is typically better than 0.1 ppm for $CO_2$ (Andrews et
al., 2014). Most tall tower sites have more than one air intake height. In this study, continuous in-situ measurements
from the highest intake are used to minimize potential influences from local sources. More information concerning
the tower sites can be found at http://www.esrl.noaa.gov/gmd/ccgg/insitu/. For the column $XCO_2$ calculation, tower
data only from 10:00-17:00 local standard time (LST) on flight days are averaged to one data point per day, as a
complement to vertical profiles within the PBL.
**2.2 Site description**
We analyze data from 19 aircraft sites and 6 tall tower sites during 2004 to 2014 (see Table S1 for a summary of site
conditions). After considering the geographic distribution of these sites in North America, we group them into eight
regions for spatial comparisons (Fig. 1). The northern west (NW) and southern west (SW) regions represent the
inflow area in the west coast of US, directly downwind of the Pacific Ocean at both higher elevations. The northern
mid-continent (NM) region represents the boreal forest and agriculture region in north-central North America. The
mid-continent (MC) region represents a dry landscape due to its high elevation (above 1.5 km on average) and semi-



arid climate. The mid-west (MW) region is strongly influenced by agriculture and temperate forest.  The southern
mid-continent (SM) represents the south-central humid temperate region, with inflow from the Gulf of Mexico
during summer.  The northeast (NE) region represents the temperate forest in north-east coast of U.S., which is
mostly downwind of regions to the west above the PBL, and downwind of its south-west regions within the PBL.
The southeast (SE) region represents the warm temperate region in the south-east coast of U.S.

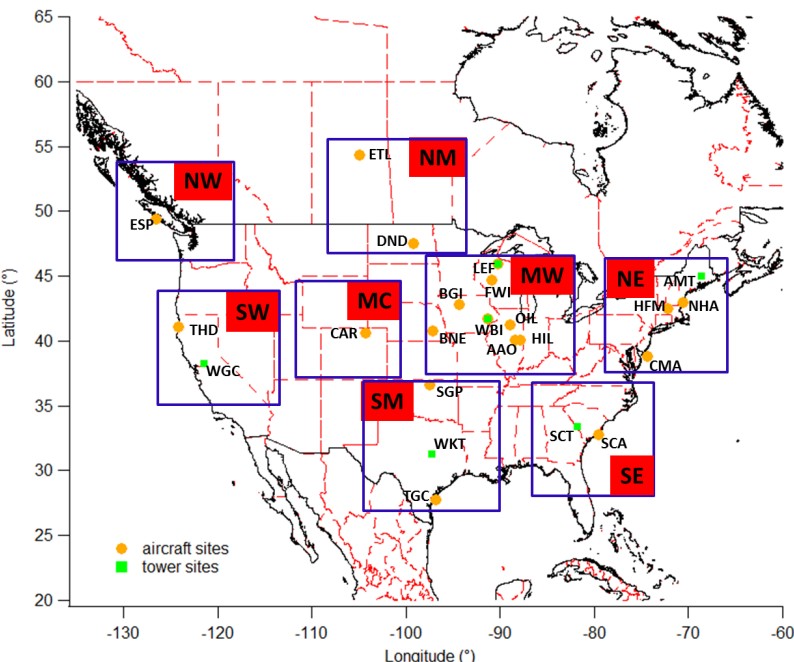


**Fig. 1.** Aircraft, tall tower, and high elevation/tower sites in the NOAA GGGRN. The eight boxes define regions
that are further discussed for spatial pattern comparison.
**2.3 Smoothing of the reference data and column XCO$_2$ calculation**
We use Mauna Loa Observatory (MLO) as a reference site.  Long-term trend of CO$_2$ measurements from this site is
removed before combining multiple years of data to calculate long-term averages. MLO is located at 19.536°N,
155.576°W, and 3397 m above sea level. Carbon dioxide measurements from this site are widely used to represent
background CO$_2$ in the Northern Hemisphere. For our study, a function consisting of a quadratic polynomial and
four harmonics is fitted to the MLO data, adopted from the method described by Thoning et al. (1989). Residuals of
the data from this function are smoothed by a low-pass filter with full-width at half-maximum in the time domain of
1.1 years. The smoothed residuals are then added back to the polynomial part of the function to produce the long-
term deseasonalized trend. This trend (see Fig. 2) is subtracted from all aircraft and tall tower measurements, as well
as from CarbonTracker model results (CarbonTracker - MLO deseasonalized trend, CarbonTracker results presented
in this study are the differences relative to observed MLO deseasonalized trend). We use 'Δ' to represent detrended





data in the following text and figures. The choice of reference site is not important for this study, since we focus on
examining the relative seasonal patterns of the detrended spatial and vertical distributions of $CO_2$ instead of the total
changes in $CO_2$ abundance attributed to global surface fluxes.

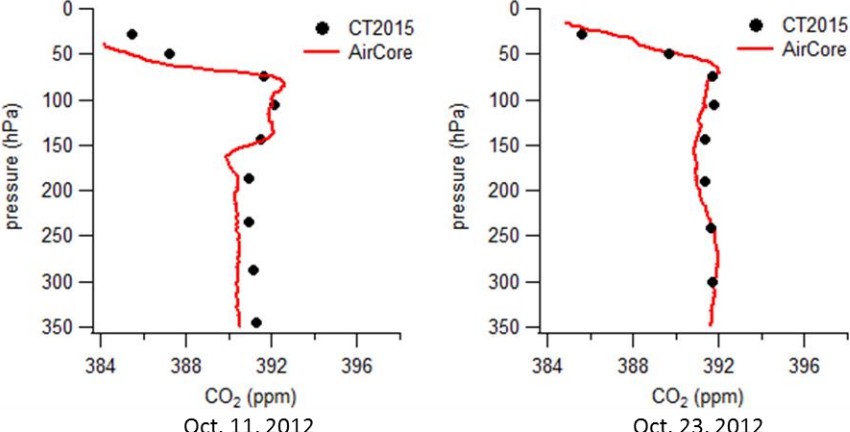

Oct. 11, 2012                          Oct. 23, 2012


**Fig. 2**. Carbon Tracker (CT2015) simulations compared with AirCore in-situ measurements in upper atmosphere.
AirCore profiles in the left and right panels are sampled near CAR and SGP, respectively.

We calculate partial column average $CO_2$ dry mole fraction using tall tower and aircraft data, and the total
column by adding simulations of high altitude $CO_2$ (above 330 hPa, ~ 8 km above sea level) from CarbonTracker.
Since geometric height from the onboard Global Positioning System (GPS) (after 2006) or inferred from the aircraft
altimeter or pressure altitude is archived with each aircraft measurement, we first convert geometric height (in
meter) to pressure (in hPa) for the pressure-weighted column $XCO_2$ calculation. This conversion uses geopotential
data from NOAA/NCEP North American Regional Reanalysis (NARR) (Mesinger et. al, 2004), available at
https://www.esrl.noaa.gov/psd/data/gridded/data.narr.html, in which the geopotential is a function of latitude,
longitude, pressure altitude and time. We interpolate the geopotential field vertically to retrieve pressure, and then
calculate dry pressure by incorporating specific humidity data from NARR. Eventually we use a trapezoidal method
to integrate over detrended vertical profiles for dry-pressure-weighted column average. For the long-term averaged
column $\Delta XCO_2$ calculation, a long-term mean vertical profile is first constructed for each month by combining 11-
year detrended data together and then average data in each 40 hPa vertical bin. To look at the long-term averaged
total column $\Delta XCO_2$ from individual aircraft sites, we combine aircraft data with upper-layer CT2015 simulations.
The NOAA CarbonTracker model assimilates $CO_2$ measurements from surface sampling networks and tall
towers to generate global 3D fields of atmospheric $CO_2$ mole fraction. The Carbon Tracker model has evolved
significantly since Peters et al. (2007). A detailed description of this model is provided in documents available at
http://carbontracker.noaa.gov. Our study utilizes CarbonTracker results from the 2015 release (CT2015), publicly
accessible at ftp://aftp.cmdl.noaa.gov/products/carbontracker/co2/CT2015/molefractions/. This version provides



$CO_2$ mole fraction over North America with $1° \times 1°$ spatial and 3 hour temporal resolutions, which are analyzed in
Sect. 3.2 and 3.3. Total column $CO_2$ calculated from CT2015 global data with $3° \times 2°$ spatial resolution is also
presented in the supporting information (SI). We have evaluated the performance of CarbonTracker in upper
atmosphere (330 to 0 hPa) by comparing its simulations with in-situ measurements from 9 AirCore profiles (Karion
et al., 2010) sampled in 2012-2014. AirCore is a ~150 m stainless steel tube that utilizes changes in ambient
pressure for passive sampling of the vertical profile. It is released using balloons and it collects a continuous sample
as it descends. It is then measured by an analyzer after it is recovered. More information about AirCore system can
also be found at https://www.esrl.noaa.gov/gmd/ccgg/aircore/. Figure 2 shows examples of AirCore profiles
compared with CT2015 in the upper atmosphere, which demonstrates good agreement. We also compare partial
column (330 to 0 hPa) averages from the 9 AirCore profiles and CT2015. Results from CT2015 agree generally well
with AirCore, with difference ranging from 0.03 to 1.22 ppm (mean value equals 0.66 ppm), which suggests that
CT2015 may have a high bias that can contribute to $0.66 \times 1/3 = 0.22$ ppm overestimate on average to the total column
average. However, AirCore is in the process of rigorous evaluation, the differences between AirCore and CT2015
are not well characterized yet since we only have a limited amount of AirCore data. It is unclear whether the
potential bias of CT2015 in this partial column is dependent on time or sampling location. Adding a constant bias
correction to all regions will not change the spatial gradients that we focus on in this study. Thus no correction is
applied when using CT2015 simulations to represent the upper 1/3 of the total column.
For uncertainty estimates, we use a 'bootstrap' method that uses random resampling and repetition of individual
vertical profiles (low bias due to high humidity was corrected), with 100 Monte Carlo runs for each column average
calculation. Uncertainty is then defined as one standard deviation of the 100 Monte Carlo results.
**3 Results and Discussions**
**3.1 Seasonal patterns and spatial gradients**
Typically one aircraft profile contains measurements at 12 different altitudes. Column $\Delta XCO_2$ can be computed for
each profile using the method described in Sect. 2.3 (Fig. S1). Figure 3 shows aircraft (at all altitudes) and tower
data (daily averages for 10:00-17:00 LST data) from all sites used in this study. Aircraft data above 2 km exhibit
much smaller seasonal variations than the full dataset, because the variations are mainly driven by $CO_2$ sources and
sinks near Earth's surface. $CO_2$ concentration is enhanced in the shallow wintertime PBL primarily due to reduced
plant photosynthesis and ecosystem respiration combined with slightly increased fossil fuel emissions. During
summer the PBL is deeper, and depletions within the PBL are due to strong terrestrial uptake that dominates over
emissions especially during June through August. During summer of 2010 to 2012, $CO_2$ from aircraft measurements
appears higher than other years in Fig.3; however, similar characteristics are not present in tower data. This
difference is due to a decrease in sampling frequency at several aircraft sites that resulted in an aliased picture of the
full summer drawdown. Since we focus on climatological mean of 11 years of data in our study, this influence is
eliminated by combining 11 years of data together into one "average year".




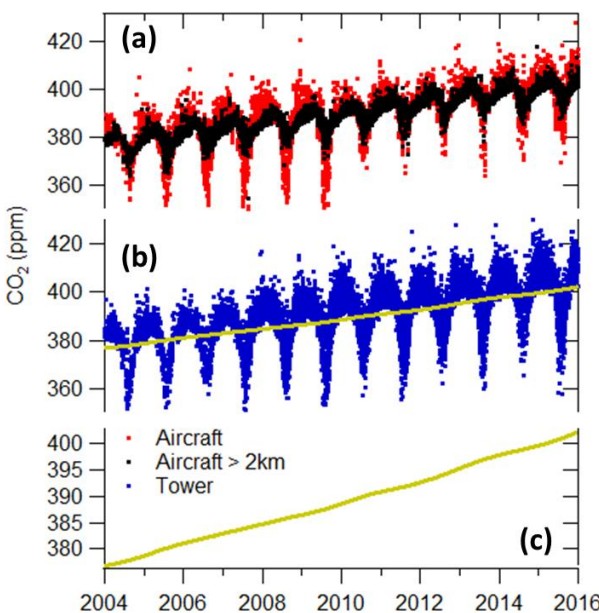

**Fig. 3.** $CO_2$ observations from aircraft (a) and towers (b). The yellow line in (b) illustrates the deseasonalized trend
at Mauna Loa (MLO), same as in (c), in which y-axis expanded.

To investigate the contributions of different altitudes to spatial gradients between regions, we divided all
measurement data into three layers according to their sampling altitudes: below 2 km, 2 - 5km, and 5 - 8.5 km masl
(Fig. 4). Smooth seasonal curves are attained from fitting data with four harmonics using the method described by
Thoning et al. (1989). The peak-to-valley amplitudes of the seasonal cycles below 2 km are the largest among the
three layers for most regions, with a minimum of 10.3 ppm in SM and a maximum of 25.0 ppm in MW. The
seasonal variation amplitudes decrease to 7.7-11.5 ppm in the 2 - 5 km layer, and further decrease to 7.2-10.0 ppm in
the 5 - 8.5 km layer. We also observe that the seasonal cycle drawdown occurs later in the layers above 2 km (see
Fig. S2, which provides similar information as Fig. 4, but seasonal curves from different vertical layers are grouped
by regions to facilitate comparisons of the phases of seasonal cycles). The seasonal $CO_2$ drawdown below 2 km is
mainly influenced by terrestrial photosynthesis and gradients are influenced by local to regional fluxes, with an
earlier onset of drawdown in southern regions than in northern regions. The seasonal cycle aloft is damped and
lagged compared to the PBL, with influences from throughout the Northern Hemisphere and with spatial gradients
likely driven by large-scale transport. The NW, SW, SM, and SE inflow regions have significant delays of more
than one month in the 2 - 5 km layer compared with the surface layer, which is likely due to the delayed phase of the
seasonal cycle in well-mixed air coming from the oceans. Vertical homogeneity of air over ocean was observed
during the HIAPER Pole-to-Pole Observations (HIPPO) aircraft campaign (Wofsy et al., 2011; Frankenberg et al.,



2016). As air masses are transported further inland, we observe reduced discrepancies of the timing of $CO_2$
drawdown between surface and upper layer air (2-5 km), which may be associated with the increased influence of
the land surface in the mid-troposphere due to strong convection over land. $CO_2$ drawdown in the 5 - 8.5 km layers
also occurs later than in the 2 - 5 km layers in most regions; however, differences between these two layers are
small. The declining amplitude and delayed phase of the seasonal cycle with altitude have been noted often (e.g.,
Tanaka et al., 1983; Ramonet et al., 2002; Gerbig et al., 2003, Sweeney et al. 2015). It demonstrates that there is lot
of important information in the vertical profile that is diminished in observations of the total column.

We find that the largest horizontal spatial gradients between regions occur below 2 km during summer time

(Fig. 4), with a maximum difference of ~15.5 ppm between MW and SM. SM and SW exhibit less pronounced
seasonal cycles, which is likely associated with air masses from the Gulf of Mexico and the Pacific Ocean,
respectively, whereas MW exhibits a deep summer drawdown partially as a result of strong regional forest and crop
uptake. Crevoisier et al. (2010) estimated the surface flux over North America using vertical $CO_2$ measurements and
average wind vectors,  and reported that annually averaged land carbon flux at the western (including SW region)
and southern regions (including SM region) were neutral. The SE region also demonstrates a less pronounced
seasonal cycle with weaker summer drawdown compared with other northern regions, which may due to the sea-
breeze influence in summer within PBL. In wintertime, $CO_2$ levels in NE and MW are higher than in other regions,
which result from regional fossil fuel and terrestrial biogenic emissions combined with transport from the west and
south.

Higher altitude data (above 2 km) exhibit only small spatial gradients. In the 2 - 5 km layer, the largest gradient

is 4 ppm in summer (Fig. 4b). It further decreases to less than 3 ppm in the 5 - 8.5 km layer (Fig. 4c).  Figure 4d
shows modeled $CO_2$ mole fractions from CT2015 for the upper troposphere and above (330 hPa to 0 hPa), which are
used to fill in above the aircraft profiles for calculation of total column $\Delta XCO_2$. Spatial gradients in this layer are
less than 0.5 ppm, suggesting that the top third of the total column has little contribution to the spatial gradients of
the total column.





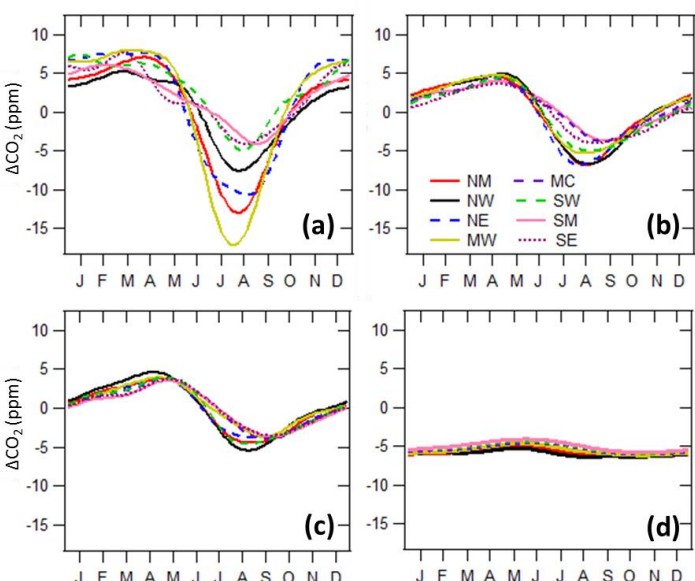

**Fig. 4.** Multi-year (2004-2014) average smooth seasonal curves of $CO_2$ relative to the long-term de-seasonalized trend at Mauna Loa for different vertical layers: (a). Aircraft and tower data under 2 km, MC is not presented because only limited data were available due to high surface elevations (>1.5 km on average) in this region; (b). Aircraft data from 2 - 5 km; (c). Aircraft data from 5 - 8.5 km; (d). CT2015 model results for layers above 330 hPa (~8.5 km) to 0 hPa (~80 km).

**3.2 Long-term mean vertical profiles**

To investigate the mean spatial gradients, we first calculate the long-term mean monthly vertical profiles as described in Sect. 2.3. In addition, each tower serves as one additional layer in the mean profile. The long-term mean tower data generally fit well in the vertical profiles from measurements of aircraft samples (Fig. 5 and Fig. 6), suggesting that the biases described in Sect. 2.1 above do not significantly affect the long-term mean. To attain profiles of the entire atmospheric column, upper layers (330 to 0 hPa) are filled in by CT2015, and the lowest data point of the measured profile is extended to ground level, defined by the mean surface elevation in that region.

Figure 5 presents two examples of long term mean profiles with data variability, which is the one standard deviation for each 40 hPa bin of aircraft data or for all flight-day tower data. Variability as large as 20 ppm is seen within the PBL in the MW region in summer, which is due to strong and heterogeneous surface vegetation uptake and ecosystem respiration combined with day-to-day changes in wind direction. All long-term mean monthly vertical profiles are presented in Fig. 6, which shows the mean temporal and vertical variability of $CO_2$ in each season, and further demonstrates the vertical propagation of seasonal $CO_2$ due to changes of surface flux. In wintertime, monotonic decrease of $CO_2$ with altitude can be observed from all regions, in which high PBL $CO_2$ is mainly driven by surface emissions and reduced vertical mixing (Denning et al., 1998; Stephens et al., 2007).




Surface $CO_2$ decreases dramatically in the growing season in those regions influenced by high plant activity, such as
NM and MW regions. For the summer vertical profiles in NE and SE region (east coast of the U.S.), the $CO_2$ mixing
ratio is elevated in the layer under 900 hPa followed by significant decreases in upper layers until 750 hPa, and then
increases with altitude until tropopause (Fig. 6). This is likely a feature of sea breeze influence. Lower-troposphere
air from the sea, lacking terrestrial uptake of $CO_2$, typically has higher $CO_2$ in summer compared with inland air.
Polluted air previously advected offshore can be brought back along with sea breeze. Without significant vertical
mixing over the marine surface, high levels of pollutants remain in those air masses. The convergence of sea breeze
with prevailing wind moving offshore may create a period with a stalled frontal structure that can aggregate air
pollutants (Banta et al., 2005). The convective internal boundary layer structure of the sea breeze system can
significantly reduce mixing height (Miller et al., 2003), and also induces higher $CO_2$ levels.  When the sea breeze is
not dominant, air advected from southwest and west (the land) can also bring in polluted air with high $CO_2$ since this
region is downwind of continental U.S. emissions (Miller et al., 2012).

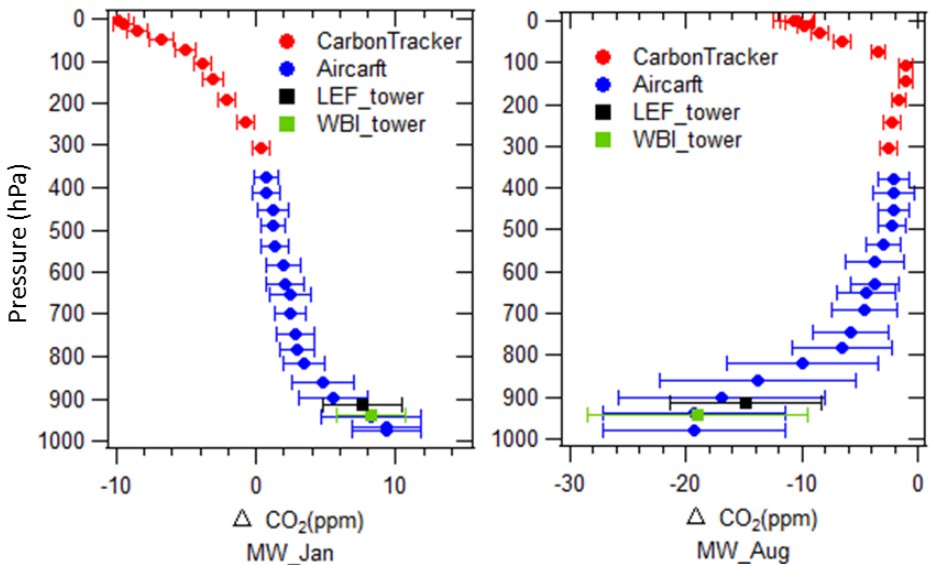


**Fig. 5**. Long-term mean (2004-2014) average vertical profiles in January (left panel) and August (right panel) in
region MW. Error bar shows one standard deviation.



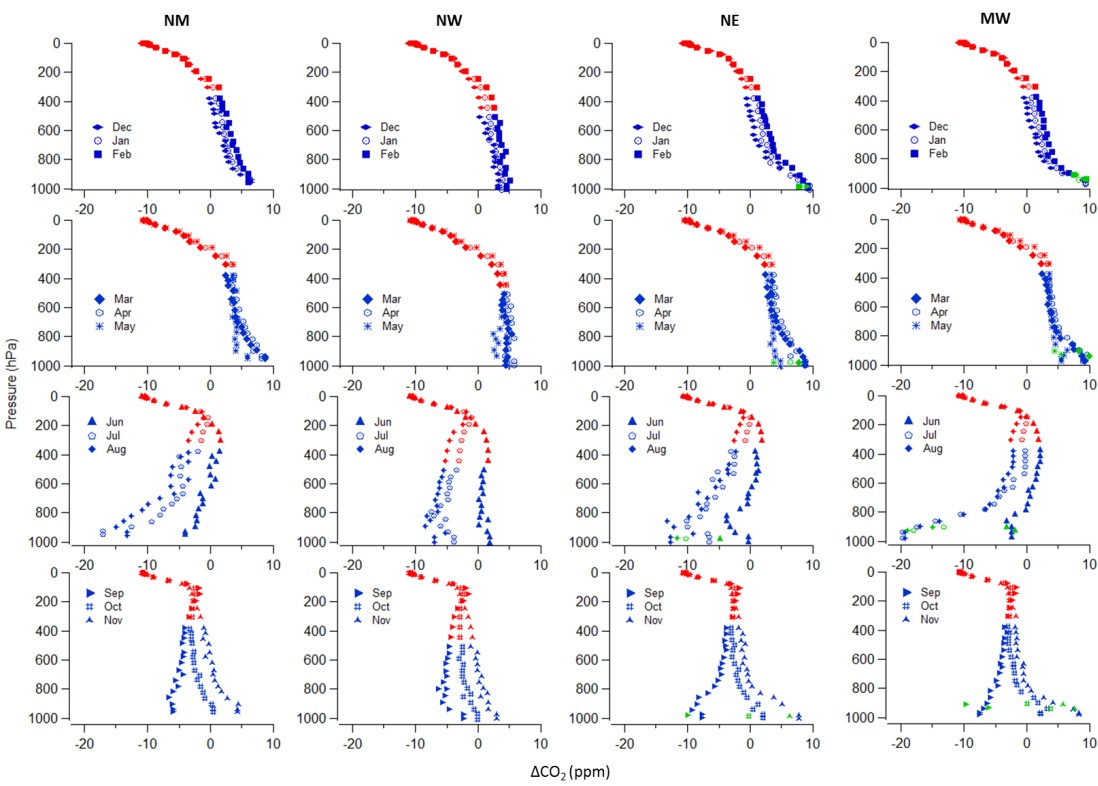


**Fig. 6a**. Long-term mean (2004-2014) monthly vertical profiles in NM, NW, NE, MW (by column, from left to
right). Blue points were calculated from observations, red points were calculated from CT2015, and green points
were calculated from tower data.




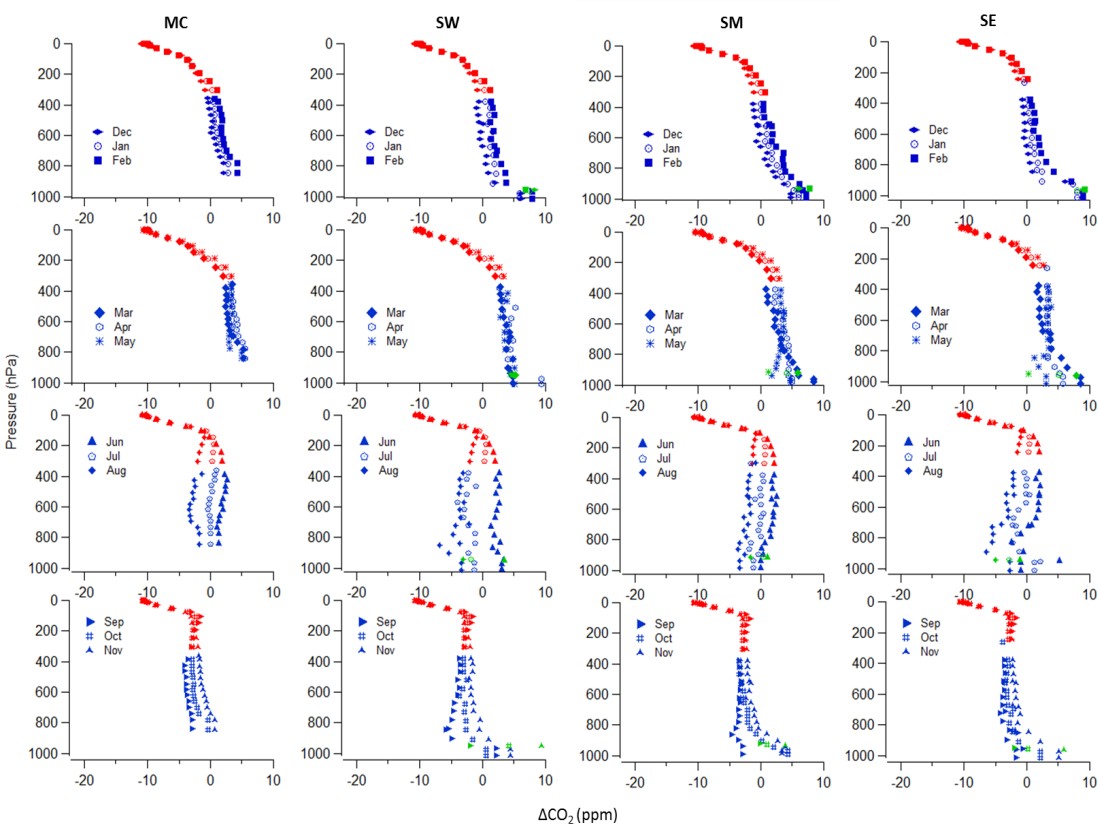


**Fig. 6b**. Long-term mean (2004-2014) monthly vertical profiles in MC, SW, SM, SE (by column, from left to right).

Blue points were calculated from observations, red points were calculated from CT2015, and green points were

calculated from tower data.






### 3.3 Partial column $\Delta XCO_2$ and total column $\Delta XCO_2$

Seasonal variations of monthly averaged partial column $\Delta XCO_2$ demonstrate maximum values in April and minimum values in August or September (Fig. 7a). The largest amplitude appears in NM, with peak-to-valley difference up to 13.5 ppm. SW, SM, SE, and MC have similar amplitudes of 7-8 ppm, smaller than other regions. To evaluate the performance of CT2015 on column $\Delta XCO_2$, CT2015 results are sampled to match the latitude, longitude, altitude and time of actual measurements. Note that aircraft profiles are not assimilated in CT2015, so aircraft data are independent of the CT2015 data assimilation. Figure 7b shows monthly partial columns of $\Delta XCO_2$ calculated from CT2015, which demonstrate good agreement with results from measurements. Only small seasonal biases exist in CT2015, with high bias occurring mostly in spring and early summer and low bias in September and October (Fig. S3). The overall differences of monthly partial column $\Delta XCO_2$ (CT2015 - measurements) mainly fall in the range of -0.64 ppm ($5^{th}$ percentile) to 0.84 ppm ($95^{th}$ percentile) with a mean difference of 0.13 ppm. These differences are of similar magnitude to the uncertainties of partial column $\Delta XCO_2$ calculated from the measurements (Fig. S4). It is clear that CT2015 captures the long-term mean variations of both phase and amplitude of partial column $XCO_2$ reasonably well.

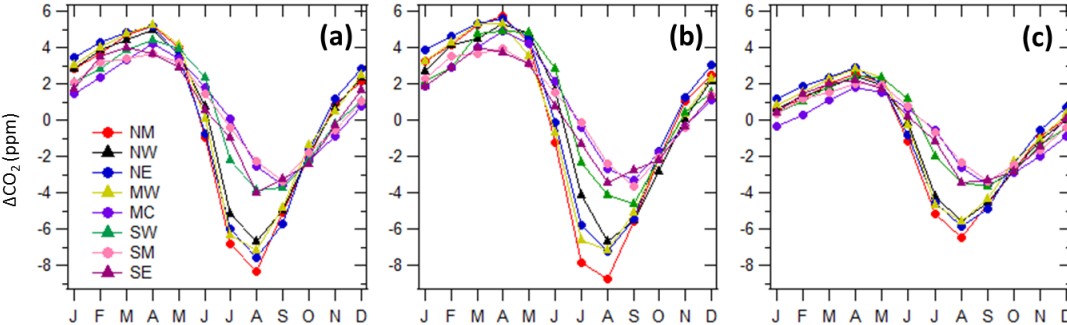

**Fig. 7**. (a). Partial column $\Delta XCO_2$ calculated from aircraft and tower data; (b). Partial column $\Delta XCO_2$ calculated from CT2015; (c). Total column $\Delta XCO_2$ calculated from aircraft and tower data and including the top layer data from CT2015.

Total column $\Delta XCO_2$ is presented in Fig. 7c. In regions NW, NM, NE, and MW, seasonal variations of total column $\Delta XCO_2$ are very similar in both phase and amplitude (8-9 ppm peak to valley). For SW, SM, SE, and MC, amplitudes are ~5.5 ppm. The smallest spatial gradients occur during May and October, which result in maximum differences among all regions of only 0.9 and 0.7 ppm, respectively. The largest spatial gradients occur during June, July and August, which result in maximum differences of 2.4, 4.5, and 4.1 ppm, respectively. It is interesting that the deepest drawdown is seen in region NM, not in region MW that encompasses the very intensive agricultural activities in the U.S. mid-west, which suggest the possibility of strong upwind influence in the NM region. The





summer drawdown of total column $\Delta XCO_2$, represented by the June to August average from CT2015, has a
magnitude that is similar to observations with differences no more than 1 ppm (Fig. 8). Based on the seasonal
patterns of total column $\Delta XCO_2$ and strength of summer drawdown, we can separate the eight regions into two
groups. The group with NW, NM, NE, and MW, has ~3 ppm stronger drawdown than the group with SW, SM, SE,
and MC. For winter total column $\Delta XCO_2$ (December to February average), the maximum spatial difference is only
1.6 ppm, with the highest total column $\Delta XCO_2$ of 1.2 ppm in NE and the lowest value of -0.3 ppm in MC.

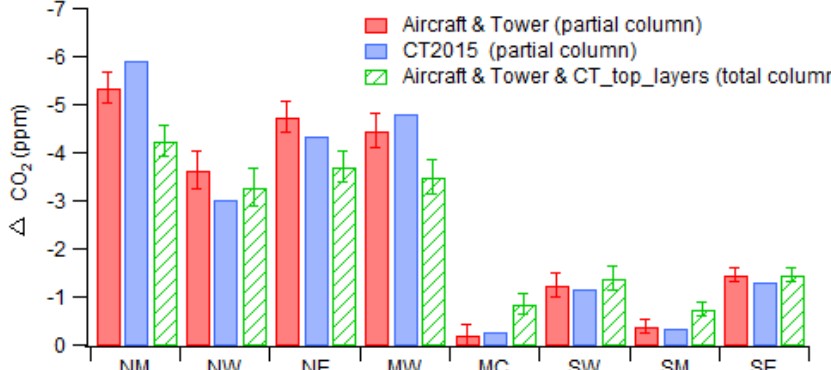


**Fig. 8**. Long-term mean (2004-2014) June to August partial and total column $\Delta XCO_2$. Error bars represent one
standard deviation from the bootstrap uncertainty calculation (see Sect. 2.3).

**3.4 Influence of large scale circulation**
Figure 9 shows long-term mean summer column $\Delta XCO_2$ calculated from CT2015, together with full column
$\Delta XCO_2$ from individual aircraft sites (note that some aircraft sites have less than 11 years of data the CT2015 shows
in Fig. 9, only aircraft sites with more than 6 years of data are presented). The fact that total column $\Delta XCO_2$ from
CT2015 agrees well with aircraft sites also supports the performance of CT2015 on a long-term average basis. The
observations show a similar summer spatial pattern, with lower column $\Delta XCO_2$ in the north and northeast regions
and higher column $\Delta XCO_2$ in the south and southwest regions (Fig. 9a). Scattered hot spots of high column $\Delta XCO_2$
associated with surface emissions from megacities, or cold spots associated with strong local uptake, are not or just
barely visible in the long-term average column $\Delta XCO_2$ map at 1°x1° resolution. Instead, the wave-like pattern of
column $\Delta XCO_2$ over North America reflects large scale circulation. To support our hypothesis on the influence of
large scale circulation, we analyze the long term mean wind pattern over North America. We can see that air masses
from northwest of the continent bring in low average column $\Delta XCO_2$, while air masses from the south (mainly the
subtropical Pacific Ocean and the Gulf of Mexico) bring in high column $\Delta XCO_2$ (Fig. 9b). The zonal gradients over
the continent, especially north of 40° N, also reflect long-term average wind patterns; southwest wind corresponds to
higher column $\Delta XCO_2$ over the western part of the continent until the wind direction shifts to west-northwest over



the eastern part of the continent. This wind pattern matches well with the geographic division of the over/under -3
ppm areas colored in green/blue in the column $\Delta XCO_2$ map (Fig. 9b). Figure 9c and 9d shows partial column
averages for free troposphere (800-330 hPa) and lower troposphere (below 800 hPa), respectively. The free
troposphere spatial gradient also demonstrates a wave-like pattern. A previous study on the total column $CO_2$ from
ground based Total Carbon Column Observation Network (TCCON) found strong correlation between the mid-
latitude column $CO_2$ and synoptic-scale variation of potential temperature ($\theta$, at 700 hPa ), a dynamic tracer for
adiabatic air transport (Keppel-Aleks et al., 2012). Thus they also propose that the variations in column $CO_2$ are
mainly driven by large-scale flux and transport.
The strong drawdown over northeast North America in summer is a consequence of long-range transport of low
$CO_2$ from northeast Eurasia, in addition to regional terrestrial uptake.  Sweeney et al. (2015) notes well-mixed
vertical profiles (up to 8 km) of $CO_2$, CO, $CH_4$, $N_2O$, and $SF_6$ from THD, ESP and PFA (Poker Flat, Alaska; 65.07$^o$,
-147.29$^o$) sites and suggests that air coming across the Pacific was strongly influenced by Asian surface fluxes
before being vertically homogenized as it passed over the Pacific Ocean. This well-mixed air forms an important
boundary condition in the column $CO_2$ of air coming into the North American continent. This was best illustrated at
sites like PFA where the summertime minimum in $CO_2$ significantly preceded maximum ecosystem uptake of $CO_2$,
implying significant influence of transported air from lower latitude regions from Asia.  We further conduct an
experiment using Carbon Tracker to investigate the importance of this effect. A control run and a "masked run" are
conducted for 2010-2012, in which the Eurasian boreal flux is turned on/off.  The MLO $CO_2$ trend from each model
scenario is used as reference background and thus removed before total column $\Delta XCO_2$ calculation. Figure 10 shows
the results for 2012 summer, which is an average summer when compared with the 2004-2014 mean pattern (Fig. 9
and Fig. 11). The maximum north-south difference reduces to ~2.5 ppm after we turn off the Eurasian boreal flux,
compared with ~5 ppm from the control run. This result combined with results from Sweeney et al. (2015)
demonstrates that the transport of low $CO_2$ resulting from large summertime Eurasian boreal uptake has a large
contribution on the overall summer total column $CO_2$ drawdown in North America.



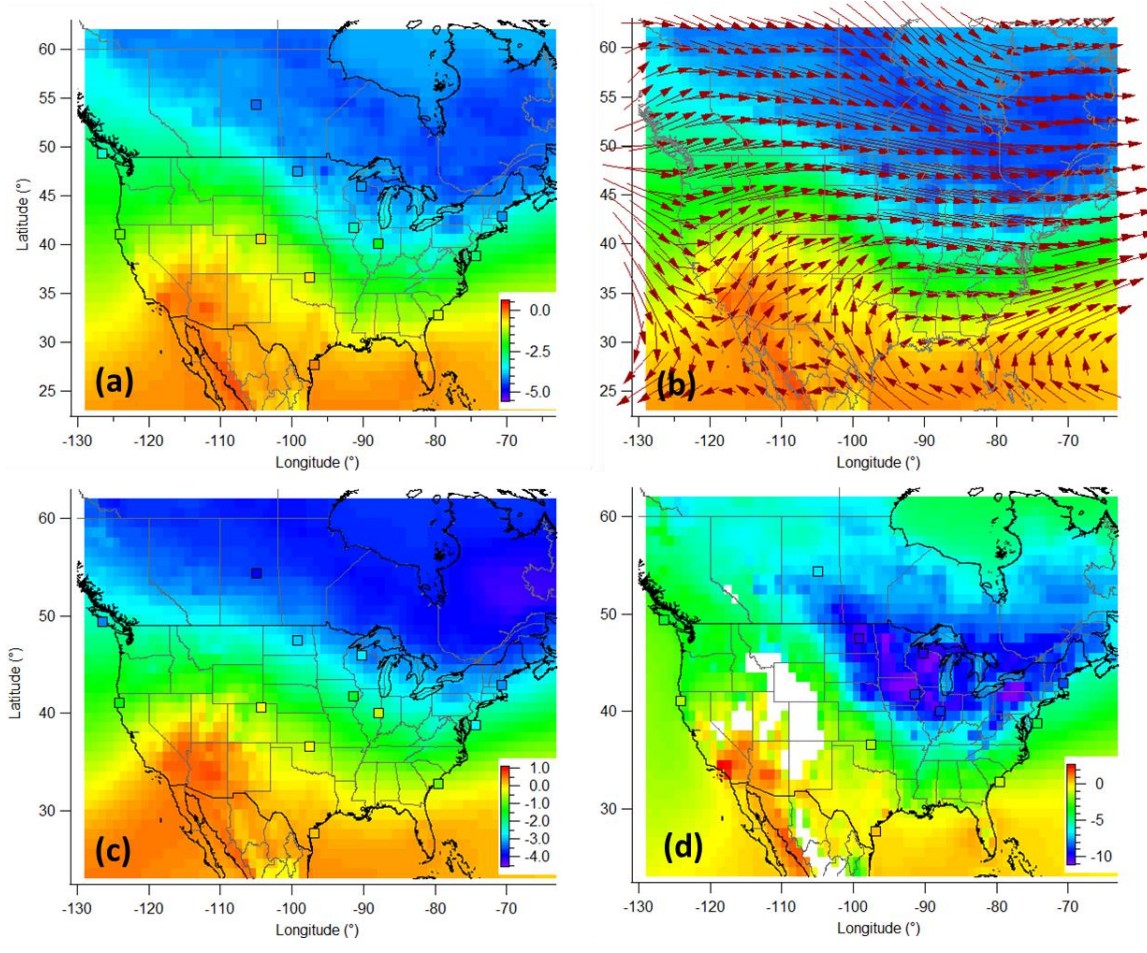


**Fig. 9**. Long-term mean (2004-2014) June-August total column $\Delta XCO_2$ from CT2015 in $1° \times 1°$ spatial resolution
with total column $\Delta XCO_2$ for 13 individual aircraft sites in squares (a), and CT2015 column $\Delta XCO_2$ overlaid with
pressure-weighted (1000 hPa to 500 hPa) mean wind vectors for the same period (b). (c) and (d) are similar as (a),
except for free troposphere (800 to 330 hPa) and lower troposphere (below 800 hPa), respectively. Note the different
color scales.



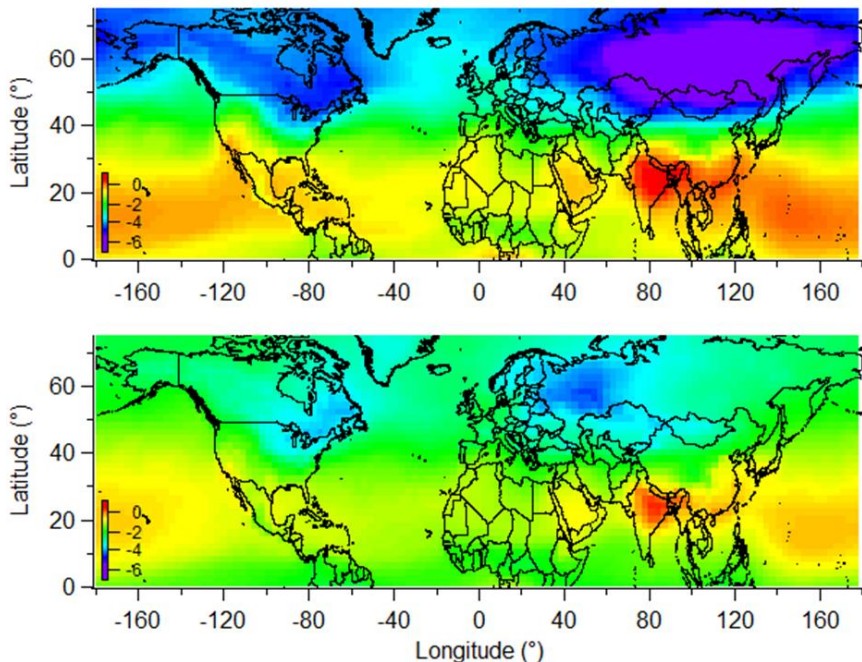

**Fig. 10**. Total column $\Delta XCO_2$ from Carbon Tracker control (top panel) and masked (bottom panel, Eurasian boreal
flux is masked) runs for 2012 June-August ($3° \times 2°$ spatial resolution). MLO trend from each individual scenario is
removed before the $\Delta XCO_2$ calculation. Same color scale is used as in Fig. 9a.

### 3.5 A comparison with apparent gradients over Europe

Figure 11 shows the climatological June - August mean modeled global column $\Delta XCO_2$ map in $3° \times 2°$ spatial
resolution, which presents smooth wave-like patterns. Reuter et al. (2014) use SCIAMACHY and GOSAT satellite
retrievals of column $CO_2$ and inverse modelling to attain surface $CO_2$ flux over European region, and suggest a large
uptake of $CO_2$ in this region. Column $\Delta XCO_2$ from CT2015 (Fig. 11) exhibits a drastically different summer spatial
pattern over Europe compared with the eight year mean (2003-2010) June through August satellite retrievals
presented by Reuter et al. (2014, their Fig. 2a). The spatial gradient from CT2015 results in a maximum 3-4 ppm
difference and a gradual pattern, instead of as much as 6 ppm from satellite retrievals. There is no sign of $XCO_2$ hot
spots from surface emissions or removals in the CT2015 spatial pattern over Europe (Fig. 11), in contrast to several
hot spots that are apparent from the 8-year averaged SCIAMACHY satellite retrievals over Ireland, U.K., northeast
of France, Belgium, Netherland, north of Germany, and south of Sweden, and low spots over the Ukraine and
Kazakhstan (Reuter et al., 2014). Although the NOAA/ESRL CT2015
(https://www.esrl.noaa.gov/gmd/ccgg/carbontracker/CT2015/) assimilates fewer observations over Europe than
Carbon Tracker Europe (http://www.carbontracker.eu/), both models produced similar fluxes over the European
region (see both websites for detailed fluxes). The $3° \times 2°$ grid from CT2015 is not likely responsible for a much



smoother pattern for Carbon Tracker, compared with the $2^o \times 2^o$ grid from satellite retrievals (Reuter et al., 2014) .
The North America region on the $3^0 \times 2^0$ grid in Fig. 11 shows similar pattern as the $1^o \times 1^o$ grid in Fig. 9, with
similar spatial difference of ~ 5 ppm. A smoother spatial distribution should be expected in Europe for the long-term
mean column $XCO_2$ (Fig. 11) due to the influences of dominating west and southwest winds in summer. Since the
satellite retrievals in Reuter et al. (2014) appear to show unrealistic column $XCO_2$ spatial gradients over Europe,
they should not be used to derive estimates of a European carbon sink. A recent study (Feng et al., 2016) using
inverse modeling suggests that satellite retrievals outside the immediate European region and a small bias of 0.5
ppm were sufficient to produce the apparent large carbon sink in the study of Reuter et al. (2014). This is expected
from elementary mass balance considerations as in Sec.1. Spatial gradients are the fundamental signals to infer
regional fluxes. Since spatial gradients from CT2015 are realistic, boreal fluxes inferred by CT2015, which shows
$0.03 \pm 2.33$ Pg C yr$^{-1}$ for Europe, should be more trustworthy.


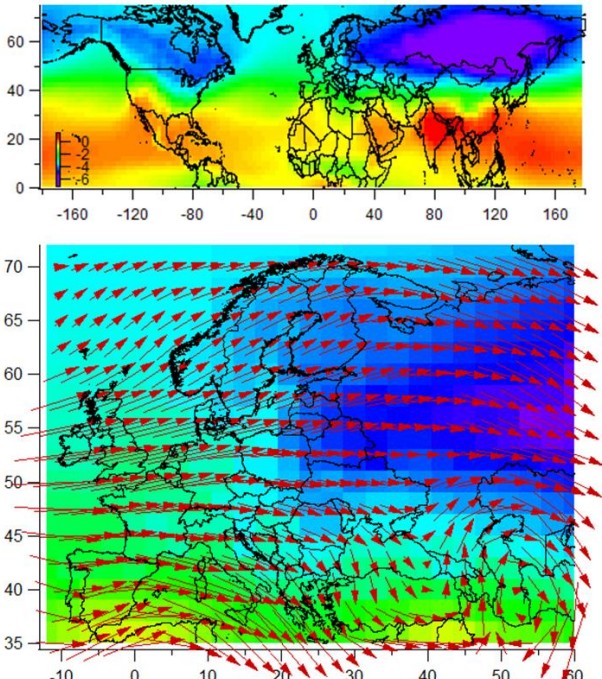


**Fig. 11**. Long-term mean (2004-2014) June - August total column $\Delta XCO_2$ from CT2015 (top panel) in $3° \times 2°$
spatial resolution, and zoom-in for Europe overlaid with pressure-weighted (1000 hPa to 500 hPa) mean wind
vectors for the same period (bottom panel). The color scale is the same as in Fig. 9a, which is scaled to reflect 6
ppm difference of $XCO_2$ to compare with satellite retrievals from Reuter et al. (their Fig. 2a, 2014).



## 4 Conclusion

Aircraft and tall tower measurements from the NOAA GGGRN provide detailed information describing the long-term average temporal and spatial variations of $CO_2$ in the PBL and the free troposphere. These data provide valuable constraints for evaluating model simulations and satellite retrievals. Seasonal cycle peak-to-peak amplitudes of $CO_2$ are largest below 2 km, where those maximum values are about twice those in the vertical layers above, indicating that most of the information on surface sources and sinks resides in the continental PBL. Large spatial gradients of $CO_2$ over North America are observed below 2 km during summer, while higher altitude data (above 2 km) have much smaller contributions to spatial gradients, with a maximum difference of only 4 ppm. The spatial differences of $CO_2$ in the upper troposphere and above (330 hPa to 0 hPa) are less than 0.5 ppm, according to CT2015. Comparison with Aircore measurements shows that the upper troposphere and lower stratospheric simulations from CT2015 are reasonably trustworthy.

Our long-term mean vertical profiles show that tower data agree well with aircraft data at similar vertical levels. Partial column $\Delta XCO_2$ was calculated from the long-term mean vertical profiles. By comparing the partial column $\Delta XCO_2$ from measurements with those from CT2015, we verify that CT2015 captures the long-term mean patterns of both phase and amplitude of partial $\Delta XCO_2$.

Large spatial gradients of $\Delta XCO_2$ only appeared in summer, during which time the north and northeast regions had ~3 ppm stronger drawdowns than the south and southwest regions. By comparing the spatial gradients of $\Delta XCO_2$ with wind vectors across North America, we find that total column $\Delta XCO_2$ patterns are equally affected by large-scale circulation patterns as by regional surface sources and sinks. A CarbonTracker experiment to investigate the impact of Eurasian long-range transport suggests that the large summer time Eurasian boreal flux contributes about half of the north-south column $\Delta XCO_2$ gradient across North America.

## Author contributions

Xin Lan was responsible for study design, data analysis, and manuscript writing. Pieter Tans was responsible for study design, data analysis, and manuscript improvement. Colm Sweeney and Arlyn Andrews provided measurement data and improved manuscript. Andrew Jacobson provided modelled data and improved manuscript. Edward Dlugokencky analyzed measurements and ensured data quality, and improved manuscript. Jonathan Kofler conducted tower measurements and improved manuscript. Molly Crotwell, Patricia Lang, and Sonja Wolter analyzed measurements and ensured data quality. Kirk Thoning provided data smoothing method.

## Acknowledgements

We especially thank John Mund for extracting NARR meteorological variables for our measurements. This research was supported by a fellowship from the National Research Council Research Associateship Programs.



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
