# Peer review of "Gradients of Column CO2 across North America from the 1 NOAA Global Greenhouse Gas Reference Network 2"

_Atmospheric Chemistry and Physics, 2017_

## Referee Comment (RC1) · Anonymous Referee #3 · 15 Jun 2017

Review of: Gradients of Column CO2 across North America from the NOAA Global Greenhouse Gas Reference Network, by Lan et al.

This paper computes total column dry-air mole fractions of CO2 (XCO2) from NOAA aircraft profiles, combined with tower measurements and capped with CT2015 stratospheric CO2. From these computed XCO2 values, they then investigate spatial gradients in 8 regions spanning most of the US and one site in Canada. The main results of this paper are that there are large north-south spatial gradients in XCO2 in summer, which are significantly influenced ($\sim$50%) by large-scale circulation patterns.

These results are interesting, and this paper approaches the total column analysis in

a new way. One particularly interesting part of this study was the "masked" run of CT which turned off the Eurasian boreal flux and reduced the North American N-S gradients by 50%. The results corroborate some of the existing literature that looks at gradients in XCO2 from ground-based remote sensing instruments that are capable of measuring CO2 in the entire column (e.g., Keppel-Aleks et al., 2011; Keppel-Aleks et al., 2012; Wunch et al., 2013).

My major comment is that this paper is begging for a direct comparison between the aircraft-derived XCO2 quantity and those measured via remote sensing at coincident TCCON stations. The obvious two sites would be at LEF (Park Falls) and SGP (Lamont) for which TCCON measurements exist since 2004 and 2008, respectively. There is also a TCCON station at ETL, but it has been measuring for less than one year and would therefore be less useful for this study. It would significantly strengthen this paper if the authors could show that their method of integrating the aircraft and extending the profile into the stratosphere with CT compares well with, or improves upon, the total column measurements from TCCON. The authors could average the TCCON XCO2 within an hour or so of the aircraft profile times for an apples-to-apples comparison. They could then use the remaining TCCON data (or at least the near-noon data) to investigate any biases or missing information caused by the relatively infrequent aircraft measurements. As part of this comparison, a more rigorous error analysis of the aircraft-derived XCO2 would be necessary.

Other comments:

L19: What does a "stronger summer drawdown" mean? Larger amplitude? Lower minimum?

L20: This sentence is contradicted by the conclusions:

"The spatial gradients of total column XCO2 across North America mainly reflect large-scale circulation patterns rather than regional surface sources and sinks."

Conclusions:

"By comparing the spatial gradients of $\Delta XCO2$ with wind vectors across North America, we find that total column $\Delta XCO2$ patterns are equally affected by large-scale circulation patterns as by regional surface sources and sinks."

Your paper seems to corroborate the sentence in your conclusions and not your abstract. Please modify the abstract accordingly.

L55: in-situ observations are sparse in global and regional coverage, and, with the exception of AirCore measurements are limited in vertical extent - most cannot measure more than 80% of the atmospheric mass.

L73: What are the uncertainties of total column XCO2 calculated from in situ measurements? According to Wunch et al. 2010, given the lack of measurements above the aircraft ceiling, the total column aircraft uncertainty is ∼0.4 ppm, which is similar to the TCCON measurement uncertainty. I would expect the errors for the profiles discussed in this paper to have larger uncertainties, since the altitude coverage for the NOAA flights is significantly smaller than the profiles used in the Wunch et al., 2010 paper.

L190: As you mention, nine AirCore profiles is inadequate to evaluate CT2015 stratospheric CO2. Perhaps you could use the other, (much) older balloon-borne or ER-2-borne stratospheric CO2 measurements (e.g., BOS (https://espoarchive.nasa.gov/archive/browse/bos/Balloon), STRAT (https://espo.nasa.gov/strat/content/STRAT_Science_Overview), ASHOE, etc.), or the more recent HIPPO and ATom aircraft profiles that often reach above the tropopause altitude, especially in wintertime.

L201: It's not clear to me why you compute the high CT bias by using the partial column comparisons. Why don't you integrate the entire CT and AirCore profiles and compare those values?

L206: This is unclear: if the stratosphere doesn't matter for your analyses, why include

it at all?

Fig 3 caption: What are the black dots?

L250: I don't understand why this sentence does not end after "It demonstrates that there is a lot of important information in the vertical profile."

Fig. 7: It seems strange that you would not include de-trended averaged TCCON XCO2 for comparison in regions that have long-term TCCON measurements (i.e., MW and SM). Additionally, Figure S1 would be an ideal place to show the SGP TCCON total column measurements.

L338: What does "deepest drawdown" mean? The lowest minimum value? The largest amplitude?

L337-339: "It is interesting that the deepest drawdown is seen in region NM, not in region MW that encompasses the very intensive agricultural activities in the U.S. mid-west, which suggest the possibility of strong upwind influence in the NM region." I agree this is interesting. Could the authors say something more about possible causes of this effect?

L343: Again, what does "strength of summer drawdown" mean here?

L373: This sentence is misleading, regarding Keppel-Aleks et al., 2012: "Thus they also propose that the variations in column CO2 are mainly driven by large-scale flux and transport." In Keppel-Aleks et al., 2012, they also state (in the abstract): "Rather than obscure the signature of surface fluxes on atmospheric CO2, these synoptic-scale variations provide useful information that can be used to reveal the meridional flux distribution."

L373: That large-scale circulation drives almost half (∼40%) of the N-S gradient in XCO2 was also shown in Wunch et al. (2013) through the interannual variability of the seasonal cycle amplitudes.

Fig 9: Why do the aircraft measurements appear to disagree significantly with CT in panel (c) in the SGP/Colorado region, and the site just south of lake Michigan?

L443: I would call 4 ppm a large difference. Please remove "only" from this sentence and quantify the "large spatial gradients" observed below 2 km.

Technical comments

L44: space after "sinks" and spurious semicolon

L90: source*s* and sinks

L154: *The* long-term trend of CO2 *measured at* this site is removed. . .

L258: . . ., which may *be* due to the sea-breeze. . .

References:

Keppel-Aleks, G., P. O. Wennberg, and T. Schneider (2011), Sources of variations in total column carbon dioxide, Atmos. Chem. Phys., 11(8), 3581–3593, doi:10.5194/acp-11-3581-2011.

Keppel-Aleks, G. et al. (2012), The imprint of surface fluxes and transport on variations in total column carbon dioxide, Biogeosciences, 9(3), 875–891, doi:10.5194/bg-9-875-2012.

Wunch, D. et al. (2010), Calibration of the total carbon column observing network using aircraft profile data, Atmos. Meas. Tech., 3(5), 1351–1362, doi:10.5194/amt-3-1351-2010.

Wunch, D. et al. (2013), The covariation of Northern Hemisphere summertime CO2 with surface temperature in boreal regions, Atmos. Chem. Phys., 13(18), 9447–9459, doi:10.5194/acp-13-9447-2013.

---

## Referee Comment (RC2) · Anonymous Referee #1 · 20 Jun 2017

Review of Lan et al. "Gradients of Column CO2 across North America from the NOAA Global Greenhouse Gas Reference Network"

This paper examines mean spatial patterns and seasonal cycles of partial and total column $CO_2$ ($XCO_2$) from NOAA aircraft locations in a number of regions across North America, and compares these to CarbonTracker estimates. They find that spatial variability of $CO_2$ is larger in certain regions as well is in summer vs. other seasons. They find that CarbonTracker generally agrees well with the aircraft-derived values for these average metrics (mesoscale and synoptic features are mostly averaged out). They find, as expected, that much (but not all) of the spatial variability is driven by remote fluxes and transport rather than local-scale fluxes. Finally, they pose a rebuttal to Reuter et al. (2014), which argues for a strong European sink based on satellite observations of column $CO_2$, arguing that the spatial patterns of $XCO_2$ derived in that work are unrealistic in comparison to CarbonTracker as well as their understanding of transport.

**Overall Comments**

In general, this paper is well-written and is a useful contribution to the literature. I have only a couple serious issues with the paper. First, since this paper is about columns, it would be useful to show direct comparisons of their aircraft-derived $XCO_2$ to TCCON, to get a sense of differences in remote sensing with respect to a similarly accurate measurement. Second, sometimes they seem to argue that nearly all spatial variability in XCO2 comes from large-scale + remote fluxes and transport, rather than regional-scale fluxes. Other times (such as in the main body), they argue that a significant contribution comes from large-scale + remote fluxes and transport, but a significant contribution also comes from more regional-scale fluxes (within North America in this study). The latter is more in agreement with their data and specifically their removal experiment of section 3.4, so we suggest they rewrite the paper to stress that both mechanisms affect the horizontal structure of XCO2.

Though the authors discuss at length their use of CT2015 to extend the aircraft profiles from 330 hPa to the top-of-atmosphere, they don't really discuss the potential errors from the fact that the aircraft measurements don't go all the way to the surface. While some sites do sample very close to the surface (e.g. 0.2 km at SGP), other sites don't even sample as low as 1 km AGL (e.g. CAR: 2.2 km agl, HIL: 1.1 km agl). Because the concentration can change quickly near the surface, the authors need to acknowledge this source of error, and ideally estimate its potential magnitude. They could do this by taking full CT profiles for their sites and comparing the column extended their way (lowest value held constant) vs. the CT value. They can also estimate it based on tower observations for a few sites.

Finally, their section on Reuter et al. (2014, hereafter R14) doesn't fully fit in with the rest of the paper. Specifically, they show that CarbonTracker (CT) agrees well with aircraft over North America. This is not surprising, given the excellent coverage of surface sites in North America, which CT assimilates. However, this

does NOT guarantee that CT will be right everywhere, which they seem to imply throughout the paper. They need to stress that this work validates CT in North America specifically. Anywhere else, and they are merely speculating. Also, they focus on the "hot spots" in the R14 map, which are likely due to inhomogenous SCIAMACHY spatiotemporal sampling as much as anything, and may disappear in a regional inversion. Regarding the European sink, the stations that CT assimilates are all in Western Europe, and much of the controversy is really how much sink is portioned between the Europe and Eurasian Boreal Transcom regions, as discussed in both R14 and Reuter et al. (2016). Their central argument that the spatial patterns of $XCO_2$ from R14 are unphysical is qualitative at best, and, from this author's perspective, partially based on the unfortunate color scheme (rainbow) chosen by R14 which can accentuate very small spatial differences. They should tone down the language to something more like to say that the spatial patterns seem unphysical, but more work would be required to really rebut their physicality. They also need to cite Reuter et al. (2016), which gives a thorough overview of the "European sink controversy".

**Specific & Technical Comments**

L54: Suggest you add language such as "or else regional-scale biases can result" and cite Chevallier et al. (2014).

L65: While satellite retrievals of $XCO_2$ can certainly have regional biases, the Inoue et al. (2016) paper is hardly thorough. It discusses one particular retrieval (the NIES retrieval) of GOSAT data, and in fact the authors appear to cherry-pick the biases of 2.09 and 3.37 ppm they list, which come from stations with the fewest colocations (3 and 1, respectively). Most of the stations in that paper have biases less than 1ppm, consistent with their comparison to TCCON, and that of Kulawik et al (2016). The authors should give a fairer representation of Inoue et al, including their TCCON comparisons, and mention the mean biases relative to TCCON given in Kulawik et al. (2016).

L88: To the 2001 and 2004 papers they cite, they should add Chevallier et al. (2014), which more explicitly demonstrate this.

L119-125: Regarding the recently-discovered high humidity bias. To me, this seems like a pretty large error, especially if the mean / typical correction is 0.5 ppm! Since this is the first paper to discuss it, a plot seems warranted – authors should plot the estimated error based on humidity, and justify this 1.7% humidity cut-off value they give. Also, the authors aren't entirely clear on the units of "humidity" here. Presumably they mean volumetric mixing ratio = dry air mole fraction; please be more clear, or use a more typical unit (such as mass mixing ratio or specific humidity).

L158: I think the authors mean Figure 3, not 2.

L277 – Here is where the authors should discuss potential errors by extending the lowest measurement to the surface (see discussion above in *General Comments*).

L308 – Please remind the reader here what "partial column" means. I think surface to 330 hPa, but it's not entirely clear.

L318 – Suggest you change the sentence to stress that CT captures these features of XCO2 *over North America* reasonably well. There is absolutely no guarantee that this will be true elsewhere, as discussed in *General Comments*. You can speculate that it will work in other locations, but you don't show it, so as written this statement is not justified.

L329 – Please define "drawdown". Does it mean peak to trough of CO2 in the seasonal cycle, or the minimum CO2 relative to MLO? The former is more typically accepted as the definition of drawdown. Then make Fig 8 consistent with your chosen definition.

L346 – Again emphasize that the agreement is only shown for North America.

L348-350 Regarding CT not showing hotspots. This is not necessarily surprising and could be due to CT's limited resolution. I've attached an unofficial plot of the CAMS nrt XCO2 over North America, for the June-August 2016 average (see e.g. Massart et al. 2016 for details), which has a resolution of roughly 0.25°, and hotspots over L.A. and Mexico city are clearly visible, consistent with the findings of Kort et al. (2012) and others.

[Figure]

L374-6 : This implies that local/regional fluxes account for ~50% of the horizontal gradient in this case.  Authors should mention this fact, rather than argue that virtually all of the spatial gradients come from large scale transport, which is what they state in the abstract.

L392: Reuter et al (2014) doesn't only use SCIAMACHY; they use GOSAT as well, and in fact, they use a number of different GOSAT retrievals, not just one, and get consistent results from all of them.  Statements in this paper should reflect this.

L409: The Feng et al. (2016) study is only applicable to global inversions.  R14 uses a regional inversion with only observations in Europe, to make themselves insensitive to an overall regional bias, and only use the gradients in the satellite data to infer the European sink.   This is somewhat at odds with the Feng et al. study, which shows that much of the apparent European sink comes from satellite observations outside of Europe.

**References**
Chevallier, F., Palmer, P. I., Feng, L., Boesch, H., O'Dell, C. W., & Bousquet, P. (2014). Toward robust and consistent regional CO2 flux estimates from in situ and spaceborne measurements of atmospheric CO2. *Geophysical Research Letters*, *41*(3), 1065-1070.

Massart, S., Agustí-Panareda, A., Heymann, J., Buchwitz, M., Chevallier, F., Reuter, M., ... & Hase, F. (2016). Ability of the 4-D-Var analysis of the GOSAT BESD XCO 2 retrievals to characterize atmospheric CO 2 at large and synoptic scales. *Atmospheric Chemistry and Physics*, *16*(3), 1653-1671.

Kort, E. A., Frankenberg, C., Miller, C. E., & Oda, T. (2012). Space‑based observations of megacity carbon dioxide. *Geophysical Research Letters*, *39*(17).

Kulawik, S., Wunch, D., O'Dell, C., Frankenberg, C., Reuter, M., Oda, T., ... & Miller, C. E. (2016). Consistent evaluation of ACOS-GOSAT, BESD-SCIAMACHY, CarbonTracker, and MACC through comparisons to TCCON. *Atmospheric Measurement Techniques*, *9*(2), 683-709.

Reuter, M., Buchwitz, M., Hilker, M., Heymann, J., Bovensmann, H., Burrows, J. P., ... & Ciais, P. (2016). How much CO2 is taken up by the European terrestrial biosphere?. *Bulletin of the American Meteorological Society*, (2016).

---

## Referee Comment (RC3) · Anonymous Referee #2 · 21 Jun 2017

Review of "Gradients of Column CO2 across North America from theÂăNOAA Global Greenhouse Gas Reference Network" by Lan et al

General Comments

In this paper, the authors use regional observations of vertical profiles of CO2 to demonstrate how vertical information shows the impacts of local emissions and transport on the atmospheric column of CO2. This is an extremely important article given the application of satellite data to the task of estimating regional fluxes. In general, the analysis is excellent and the conclusions are appropriate, and I recommend publication after a few changes to address the concerns below.

Throughout the paper, the expression "drawdown" is used when what the authors really mean is "negative anomalies". Âă This is an important conceptual difference, since the entire point of the paper is to determine the information content of column averages on fluxes versus transport. Given the discussion of meteorological phenomena such as the sea breeze and the resulting difficulty in interpretation of anomalies, it's important to not mislead the reader with this phrasing.

Additionally, the reader might be lead to the somewhat implicit conclusion that XCO2 can't serve as a constraint on surface fluxes, but this might be an artifact of the multi-month averaging that is used in the analysis in later sections. At least some discussion of the ability of models such as CT to capture transient features needs to be put forth, because these transient spatial gradients could indeed be attributed to fluxes given sufficient accuracy and adequate transport modeling, as has been shown in numerous OSSEs (Liu et al [2014], Miller et al [2007], Rayner and O'Brien [2001], to name just a few). Connecting the analysis in this paper to those earlier studies is critical to readers trying to assess the conclusions of inversion work with OCO-2 and GOSAT.

Specific comments:

Figure 2: Are there statistics of goodness of fit? ÂăThese two examples may not be representative.

Line 207: Can you explain the word "random" here? ÂăIf all of the profiles are given the same weight in the sampling distribution, then this isn't really a measure of uncertainty, but rather a weighted standard deviation that would be extremely sensitive to the 100 particles you selected. ÂăIf there were a "prior" uncertainty placed on each profile, how was that done?

Line 216: ÂăThere is also some increase due to the shallow PBL alone. ÂăIt would be good to know what fraction is from the enhancement is due just to boundary layer dynamics. ÂăSimilarly for the summer.

Line 228: The PBL height is different at different locations, and through different seasons. ÂăHow much is the chosen division of the atmosphere mis-attributing boundary layer into the free troposphere, and vice versa? ÂăThis might be a small detail, but it does impact the conclusions later about the seasonal strength of sources and sinks by region.

Line 257: "The SE region also demonstrates a less pronounced seasonal cycle with weaker summer drawdown compared with other northern regions, which may due to the sea-breeze influence in summer within PBL." Is this a statement about the actual impact of the sea breeze on the fluxes, or is it an assertion that we can't interpret the column due to the meteorology?

Figure 4: It would be useful to have a fifth panel that shows full column XCO2 with the CT extension here, to re-inforce the assertion on line 249-250 about the information lost by considering the total column.Âă

Figure 9a: The multi-month average XCO2 gradients can easily miss transient features that could, in theory, be well captured by a regional transport model having spatial resolution that is sufficient to capture synoptic features such as fronts. These features could be attributed to fluxes under this assumption, provided biases are small. That doesn't discount this analysis, but it does imply the need to make assertions about the constraint of XCO2 on surface fluxes.

Fig 9b: This vertically averaged wind vector plot doesn't really match the spatial gradient well, in some cases actually being perpendicular to the field. ÂăIt would make more sense to use the potential temperature at 700mb, as did Keppel-Aleks et al in the reference you cite. ÂăAlternately, the 500mb geopotential height is a commonly used field for synoptic scale transport in NWP.

Figure 10: Can you show the same plots, but for the partial columns that are depicted in Figure 9c and d? That would really drive home the point about transport versus local fluxes.

[Figure]

Discussion of Reuter et al [2014]: Is it possible that the differences are due to the manner in which the anomalies are computed? Or are you asserting that the gradients are due to satellite measurement bias? Stating that they "should not be used" needs a bit more justification here.
* * *

---

## Author Response (AR1)

Reply to Reviewer

Thank you for reviewing our paper.

General comments:

1. Throughout the paper, the expression "drawdown" is used when what the authors really mean is "negative anomalies".

"Drawdown" refers to the variation amplitude in the seasonal cycle that mostly resulted from strong plants activity. We will clarify this point in the revised paper.

2. Additionally, the reader might be lead to the somewhat implicit conclusion that XCO2 can't serve as a constraint on surface fluxes, but this might be an artifact of the multimonth averaging that is used in the analysis in later sections. At least some discussion of the ability of models such as CT to capture transient features needs to be put forth, because these transient spatial gradients could indeed be attributed to fluxes given sufficient accuracy and adequate transport modeling, as has been shown in numerous OSSEs (Liu et al [2014], Miller et al [2007], Rayner and O'Brien [2001], to name just a few). Connecting the analysis in this paper to those earlier studies is critical to readers trying to assess the conclusions of inversion work with OCO-2 and GOSAT.

We agree with the reviewer that transient spatial gradients could indeed be attributed to fluxes. The ability to capture transient feature is important criteria for modeling performance. Transient features, for example, the seasonal cycle, are generally easier to model since the signals are much larger compared with annual average. However, very small biases in seasonal cycle can still cause drastically biased annual fluxes and unrealistic compensating fluxes. The ability to capture transient features and perfect transport in modelling are not enough. We think it is critical to reduce biases in measurements as inputs for inversions. Previous study by Masarie et al. (2011) had evaluated the impact of CO2 measurement bias on CarbonTracker flux estimates and found that 1 ppm bias at one site, the Park Fall ,Wisconsin (LEF site in our study), can cause 68 Tg C/yr bias in flux estimate for

Temperate North America (~ 10% of the estimated North American annual terrestrial uptake). Flux estimate errors are also found in Europe and boreal Eurasia to compensate for the errors in North America.

Whether or not the column CO2 can serve as a good constraint on surface fluxes really depends on the biases in column CO2 retrievals. High accuracy is needed from column CO2 product to be useful to constrain surface fluxes because column CO2 is a total column average, thus it is not as sensitive to surface fluxes as surface measurements. For example, a simple mass balance argument shows that all U.S. CO2 emissions from fossil fuel burning (~1.4 Pg yr-1) create a total column enhancement of only **0.6 ppm** on average in air parcels over the East Coast compared to the West Coast and Gulf Coast if we assume a residence time of the emissions of 5 days to pass the contiguous U.S. (~8×10$^{12}$ m2). From the noise to signal ratio perspective, it is important to have high accuracy in column CO2 products. Considering current state of biases in remote sensing products, we think extensive works are needed to reduce biases. This can be done given enough well-calibrated surface measurements and vertical profiles.

Our study provides an approach to check whether column CO2 retrievals are satisfying the high accuracy requirement on some levels. Since spatial gradient is the core of deriving reliable regional fluxes, we look into the spatial gradient in long-term multimonth averages, which should be easier for models and column CO2products to achieve compared with some short-term metrics (e.g. diurnal cycles).

Specific comments:

1. Figure 2: Are there statistics of goodness of fit? These two examples may not be representative.

We have 9 independent AirCore profiles during study period, which are all sampled nearby CAR and SGP sites. That's why we show one sample from each site. We will clarify this point in the revised paper. The flowing figures show AirCore vs CT2015 modelled $CO_2$ above

330hPa. The left panel shows direct point to point comparisons from all 9 profiles at the vertical levels of CT2015; the right panel shows partial column (0- 330hpa) averages. Due to the limited amount of AirCore profiles, we prefer not to put too much discussion of estimate uncertainty in our paper. In addition, the upper 1/3 of the atmosphere is not important when we are looking at long-term averaged spatial gradients in total column because there is little spatial variability in the upper atmosphere. We can see the fit is generally good even when we are comparing at profile by profile basis without temporal averaging.

[Figure]

2. Line 207: Can you explain the word "random" here? If all of the profiles are given the same weight in the sampling distribution, then this isn't really a measure of uncertainty, but rather a weighted standard deviation that would be extremely sensitive to the 100 particles you selected. If there were a "prior" uncertainty placed on each profile, how was that done?

 'random' means all of the profiles are given the same weight in the Monte Carlo resampling, and the resulting 100 column averages values have a relatively large range since this value are sensitive to the combination of  the profiles. That is what we want to use to represent the uncertainty, to account for the atmospheric variability without assuming a giving distribution of the vertical profiles. A 'prior' uncertainty is likely to represent mostly the measurement uncertainty, which is too small without fully consider the atmospheric variability.

3.  Line 216: There is also some increase due to the shallow PBL alone. It would be good to know what fraction is from the enhancement is due just to boundary layer dynamics. Similarly for the summer.

We agree with the reviewer. However, putting a number on the fractions of enhancement from either changes of PBL or flux requires model with good PBL dynamics and reasonable flux estimates. Good PBL simulations are still very challenging for models. We think it is beyond the scope of this study.

4.  Line 228: The PBL height is different at different locations, and through different seasons. How much is the chosen division of the atmosphere mis-attributing boundary layer into the free troposphere, and vice versa? This might be a small detail, but it does impact the conclusions later about the seasonal strength of sources and sinks by region.

For the < 2 km measurements, we believe both PBL and fluxes are important driving factors for the signals we observe. By choose a giving height (2 km), we have removed some of the PBL effect by compensating the PBL with some free-troposphere air. If we use actual temporally changing PBL height as threshold, we will see stronger influences of PBL in wintertime as the $CO_2$ levels in shallow PBL are even higher compared to our approach. At this moment, we cannot estimate the total influence of PBL and completely remove it without using a model, which may also have big uncertainty in PBL estimates. Our study does not focus on the discussion of the actual number of the fluxes.

5.  Line 257: "The SE region also demonstrates a less pronounced seasonal cycle with weaker summer drawdown compared with other northern regions, which may due to the sea-breeze influence in summer within PBL." Is this a statement about the actual impact of the sea breeze on the fluxes, or is it an assertion that we can't interpret the column due to the meteorology?

We state the possible impact of sea breeze on the data and the gradient without implying the interpretation of column data. Given a model with good performance on sea-land breeze and sufficient accuracy in column CO2, we should be able to interpret the column data that influenced by sea-land breeze. The message here is that we should take into account of the sea-land breeze effect when interpreting the data.

6. Figure 4: It would be useful to have a fifth panel that shows full column XCO2 with the CT extension here, to re-inforce the assertion on line 249-250 about the information lost by considering the total column.

It is presented in Figure 7c. We will point the reader to Figure 7c in this part of the description.

7. Figure 9a: The multi-month average XCO2 gradients can easily miss transient features that could, in theory, be well captured by a regional transport model having spatial resolution that is sufficient to capture synoptic features such as fronts. These features could be attributed to fluxes under this assumption, provided biases are small. That doesn't discount this analysis, but it does imply the need to make assertions about the constraint of XCO2 on surface fluxes.

The purpose of Figure 9a is to show that CT2015 can compare well with XCO2 from aircrafts data at this temporally averaged scale, and we should expect a smooth spatial gradient pattern at this temporal scale. This is another baseline to evaluate the performances of models and column CO2 retrievals, in additional to transient features. Please also see our responses in General Comment.

8. Fig 9b: This vertically averaged wind vector plot doesn't really match the spatial gradient well, in some cases actually being perpendicular to the field. It would make more sense to use the potential temperature at 700mb, as did Keppel-Aleks et al in the reference you cite. Alternately, the 500mb geopotential height is a commonly used field for synoptic scale transport in NWP.

We agree with the review that potential temperature and geopotential height patterns can better match with CO2 spatial gradient; however, these terms are less straight forward when we interpretation the transport and point to the source regions. We think the wind pattern is sufficient to show the upwind locations we discuss in this study.

9. Figure 10: Can you show the same plots, but for the partial columns that are depicted in Figure 9c and d? That would really drive home the point about transport versus local fluxes.

We will provide these figures in the revised paper:

[Figure]

**Figure.** Total and partial column ΔXCO2 from Carbon Tracker control (left panels: (a), (b), and (c)) and masked (right panels, (d), (e), and (f). Eurasian boreal flux is masked) runs for 2012 June-August (3° × 2° spatial resolution). MLO trend from each individual scenario is removed before the ΔXCO2 calculation. (a) and (d) show total column averages. (b) and (e) show partial column averages for free troposphere (800 hPa to 330 hPa). (c) and (f) show partial column averages for lower troposphere (below 800 hPa). (a) and (d) use the same color scale as in Fig. 9a., which reflects maximum 6 ppm gradient. Color scale in (b) and (e)

also shows maximum 6 ppm gradients for comparison with (a) and (d); however, the actual values are different. Color scales in (c) and (f) are larger to reflect large spatial gradients in lower troposphere.

10. Discussion of Reuter et al [2014]: Is it possible that the differences are due to the manner in which the anomalies are computed? Or are you asserting that the gradients are due to satellite measurement bias? Stating that they "should not be used" needs a bit more justification here.

We don't think the manner in which the anomalies are computed is the reason for the unrealistic gradient. According to their description, the long-term trend has been removed when computed the anomalies, just as in our study. They use 8-year summertime data, which should provide enough data for a reasonable averaged pattern. Thus the quality of the retrievals is likely the reason for an unrealistic pattern. Since the spatial gradient is the core of inversion study to estimate fluxes, it is critical for the input data to have good spatial gradients. We will add in more discussion in the revised paper.

Reply to Reviewer:

Thank you for reviewing our paper.

General comments:

My major comment is that this paper is begging for a direct comparison between the aircraft-derived XCO2 quantity and those measured via remote sensing at coincident TCCON stations. The obvious two sites would be at LEF (Park Falls) and SGP (Lamont) for which TCCON measurements exist since 2004 and 2008, respectively. There is also a TCCON station at ETL, but it has been measuring for less than one year and would therefore be less useful for this study. It would significantly strengthen this paper if the authors could show that their method of integrating the aircraft and extending the profile into the stratosphere with CT compares well with, or improves upon, the total column measurements from TCCON. The authors could average the TCCON XCO2 within an hour or so of the aircraft profile times for an apples-to-apples comparison. They could then use the remaining TCCON data (or at least the near-noon data) to investigate any biases or missing information caused by the relatively infrequent aircraft measurements. As part of this comparison, a more rigorous error analysis of the aircraft-derived XCO2 would be necessary.

We agree with the reviewer that it will be interesting to compare TCCON with aircraft + CT based total column CO2. The following figure can give us some ideas about this comparison. TCCON values in this figure are daily averages computed from mid-day data (10 to 15 LST). Outliers in TCCON data are filtered out, by using 3.S.D threshold of residuals (with iteration) from a quadratic fit on data for each day.

[Figure]

TCCON Data DOI       10.14291/tccon.ggg2014.lamont01.R0/1149159

TCCON reference      Wunch, D., G. C. Toon, J.-F. L. Blavier, R. A. Washenfelder, J. Notholt, B. J. Connor, D. W. T. Griffith, V. Sherlock, and P. O. Wennberg (2011), The total carbon column observing network, Philosophical Transactions of the Royal Society - Series A: Mathematical, Physical and Engineering Sciences, 369(1943), 2087-2112, doi:10.1098/rsta.2010.0240. Available from: http://dx.doi.org/10.1098/rsta.2010.0240

However, it is not an apple-to-apple comparison between our aircraft-based column $CO_2$ and TCCON. The sampling time and location are different. Aircraft sampling typically takes ~ 30 min to get the whole profile through downward spiral, mostly within $0.1^o$ of the site location. For TCCON that sampling the whole column instantly at an angle, the actual sampling location, especially at high altitudes, may be far away from our aircraft inlets even though the distance on the ground is much smaller. Mismatches caused by both sampling time and location should be examined carefully if we want answers for accuracy, which should be the topic of another in-depth study. In addition, the aircraft and tower measurements are calibrated on WMO scale; however, direct calibrations of remote sensing instruments are not available. It makes more sense to compare TCCON against calibrated measurements for accuracy instead of the other way around. Our study focuses on the long-term mean spatial patterns of column $CO_2$ over North America and the influence of large-scale transport; we don't think analysis on two individual TCCON sites can provide extra information regarding to our goal.  Previous study by Keppel-Aleks et al. (2012) already provides analysis on TCCON data and meridional spatial patterns.

Specific comments:

1.      L19: What does a "stronger summer drawdown" mean? Larger amplitude? Lower minimum?

"Drawdown" refers to the variation amplitudes in the seasonal $CO_2$ signals that mostly resulted from strong plants activity. We clarify this point in the revised paper.

2.      L20: This sentence is contradicted by the conclusions:

"The spatial gradients of total column XCO2 across North America mainly reflect largescale circulation patterns rather than regional surface sources and sinks."

Conclusions:

"By comparing the spatial gradients of XCO2 with wind vectors across North America, we find that total column XCO2 patterns are equally affected by large-scale circulation patterns as by regional surface sources and sinks."

Your paper seems to corroborate the sentence in your conclusions and not your abstract.

Please modify the abstract accordingly.

We do not mean that local fluxes have no influence on long-term averaged column CO2 spatial pattern. But the major contributor is transport, instead of surface flux. We have results from CT experiments showing that 50% of North-South gradient are due to Eurasia flux, which is stated in both abstract and conclusion parts. Since Eurasia flux is not the only transport signals we observed (upwind Canadian region is likely contribute to transport signal too), we actually expected larger than 50% transport influence in total.  We will clarify the statement in both abstract and conclusion.

3.      L55: in-situ observations are sparse in global and regional coverage, and, with the exception of AirCore measurements are limited in vertical extent - most cannot measure more than 80% of the atmospheric mass.

We agree with the reviewer that aircraft measurements have vertical limitations. However, we should also acknowledge that most of the flux signals reside in the lower part of the atmosphere. There is little spatial variation of atmospheric CO2 in the stratosphere over mid-latitude region (Andrews et al., 2001). Thus this part of the atmosphere has little contribution to the spatial patterns of total column CO2 that are used to retrieve surface sources and sinks.

   (Citation: Andrews, A., Boering, K., Daube, B., Wofsy, S., Loewenstein, M.,Jost, H., Podolske, J., Webster, C., Herman, R., Scott, D., et al.: Mean ages of stratospheric air derived from in situ observations of CO2, CH4, and N2O, J. Geophys. Res. Atmos., 106, 32295–32314, 2001)

4.      L73: What are the uncertainties of total column XCO2 calculated from in situ measurements? According to Wunch et al. 2010, given the lack of measurements above the aircraft ceiling, the total column aircraft uncertainty is 0.4 ppm, which is similar to the TCCON measurement uncertainty. I would expect the errors for the profiles discussed in this paper to have larger uncertainties, since the altitude coverage for the NOAA flights is significantly smaller than the profiles used in the Wunch et al., 2010 paper.

We have provided the uncertainty in partial column $CO_2$ derived from tower and aircraft measurements in SI, figure S4, which ranges from 0.12 to 0.96 ppm with average value of 0.32 ppm over all eight regions for long-term monthly averaged values (uncertainty in summer months are higher). Uncertainty in vertical profile mainly reflects atmospheric variability instead of just instrument errors. In our study, several regions have more than one aircraft sites. When we are averaging over those sites over a total of 11-year vertical profiles, the uncertainty goes down since we have reduced atmospheric variability significantly.   The uncertainty in Wunch et al.,2010 is based on individual profile without long time averaging, thus it is not the same as in our study .

We cannot properly assign an exact number as the uncertainty for the top 1/3 of the atmosphere without routine in-situ measurements.  However, spatial gradient and atmospheric variability is very small in this part of the atmosphere, and thus it is not important in our study.

5.      L190: As you mention, nine AirCore profiles is inadequate to evaluate CT2015 stratospheric CO2. Perhaps you could use the other, (much) older balloon-borne or ER-2-borne stratospheric CO2 measurements (e.g.,BOS (https://espoarchive.nasa.gov/archive/browse/bos/Balloon), STRAT (https://espo.nasa.gov/strat/content/STRAT_Science_Overview), ASHOE, etc.), or the more recent HIPPO and ATom aircraft profiles that often reach above the tropopause altitude, especially in wintertime.

Stratospheric CO2 has very small spatial gradient (Andrews et al., 2001) that contribute to total spatial gradient of total column CO2. Thus it is not important for our study. We only need reasonable stratospheric CO2 results to compensate the missing column information of stratospheric CO2. We could've simply expand the partial column results to the total column since we know we have ~ 70% of the column with most of the spatial gradient signals; however, we opt to use the CT modelled stratosphere CO2 to account for the effect of slightly delayed seasonal cycles in stratosphere.

6.      L201: It's not clear to me why you compute the high CT bias by using the partial column comparisons. Why don't you integrate the entire CT and AirCore profiles and compare those values?

The top 1/3 partial column is the part from CT that we use to calculate total column CO2, in addition to aircraft and tower measurements. Integrating entire vertical profiles of AirCore will introduce extra uncertainty of AirCore measurements near surface, which is mostly related to the seal time of the AirCore that is not relevant to the stratospheric part of the measurements. We have modified the sealing process for recent AirCore samples.

7.      L206: This is unclear: if the stratosphere doesn't matter for your analyses, why include it at all?

Please also see answers for question 3 and 5. We want the results for total column in addition to the partial column that ceiled at 8 km, to provide baseline spatial patterns for comparison with total column retrievals from remote sensing instruments.  For the stratosphere, we show in our study that Carbon Tracker's performance is good when compared with individual AirCore profile. For long-term averages, Carbon Tracker results are good enough for our study.

8.      Fig 3 caption: What are the black dots?

Black dots in Fig.3 shows the aircraft data above 2km. It is showed in the figure legend.

9.  L250: I don't understand why this sentence does not end after "It demonstrates that there is a lot of important information in the vertical profile."

We state "It demonstrates that there is lot of important information in the vertical profile that is diminished in observations of the total column". We think it is important conclusion in our study: most spatial gradient signals reside in lower troposphere (that's why it is important for measurements to be sensitive to this part of atmosphere). This conclusion cannot be demonstrated by column measurements.

10.  Fig. 7: It seems strange that you would not include de-trended averaged TCCON XCO2 for comparison in regions that have long-term TCCON measurements (i.e., MW and SM). Additionally, Figure S1 would be an ideal place to show the SGP TCCON total column measurements.

Please see response to general comment.

11.  L338: What does "deepest drawdown" mean? The lowest minimum value? The largest amplitude?

We mean the largest amplitude in seasonal cycles among the 8 regions. We will clarify this in the revised paper.

12.  L337-339: "It is interesting that the deepest drawdown is seen in region NM, not in region MW that encompasses the very intensive agricultural activities in the U.S. midwest, which suggest the possibility of strong upwind influence in the NM region." I agree this is interesting. Could the authors say something more about possible causes of this effect?

This is evidence that column CO2 is not dominated by surface flux. Even though the boreal uptake is the largest in MW in summer due to intense plant activities, the column CO2 does not show the lowest value in MW because transported signals have major contribution on column CO2.

13.     L343: Again, what does "strength of summer drawdown" mean here?

"strength of summer drawdown" means the variation amplitude in seasonal cycles. We will clarify this in the revised paper.

14.     L373: This sentence is misleading, regarding Keppel-Aleks et al., 2012: "Thus they also propose that the variations in column CO2 are mainly driven by large-scale flux and transport." In Keppel-Aleks et al., 2012, they also state (in the abstract): "Rather than obscure the signature of surface fluxes on atmospheric CO2, these synopticscale variations provide useful information that can be used to reveal the meridional flux distribution."

Keppel-Aleks et al.(2012) finds similar results as our study, and states in their abstract that "New observations of the vertically integrated CO2 mixing ratio <CO2> from ground-based remote sensing show that variations in <CO2> are primarily determined by large-scale flux patterns." They specifically state in sector 3.1 that "In summary, the comparison between drawdown in <CO2> and eddy covariance flux confirms that while regional information is contained in column observations, **these regional flux signals are obscured by larger-scale variations in <CO2>** even on the short timescale." We both agree with this. Since column CO2 primary reflects large-scale flux, it is logical that local signals are significantly "diluted" in the total column.

The later statement is actually referring to their method to use both potential temperature and column CO2 to estimate flux gradients (the distribution), but not the flux itself. This method is useful to evaluate biosphere models.

15.    L373: That large-scale circulation drives almost half (~40%) of the N-S gradient in XCO2 was also shown in Wunch et al. (2013) through the interannual variability of the seasonal cycle amplitudes.

We will cite Wunch et al. (2013) in revised paper. However, we should also acknowledge that interannual variability is not the same statistic metric as the long-term averaged spatial patterns we discuss in our study.  We should expect that smaller scale features (local/regional fluxes) are further dampened in long-term time scale.

16.    Fig 9: Why do the aircraft measurements appear to disagree significantly with CT in panel (c) in the SGP/Colorado region, and the site just south of lake Michigan?

CT in panel (c) shows the partial column $CO_2$ between 800 hpa to 330 hpa. For SGP and CAR (-0.30 and -0.23 ppm, respectively, based on aircraft measurements), CT results show -0.74 and -0.57 ppm (these numbers are provided in the SI in the revised paper). The mismatches are within ~0.4 ppm which is similar to our uncertainty levels. There are a few possible reasons for these mismatches. The aircraft is designed to sample $0.1° \times 0.1°$ area, while CT is $1° \times 1°$ spatial resolution. Thus CT may show smoother results. In this case, CT compared better with aircraft measurements in background sites than those influenced by local anthropogenic emissions significantly. SGP is highly polluted by Oil and Gas operation. CAR is also influenced by ONG operations; however, the mountain terrain is challenge for the model too.

For the site HIL (south of Lake Michigan), we have recently found unrealistic high CO2 levels from some profiles that might associated with cabin air contamination. As a precaution, we have removed this part of data (including both 2012 and 2013 summer) in our study after comparing mid-troposphere measurements with the other 4 aircraft sites (LEF, WBI, CAR, and DND) around. We have updated relevant results (including figures) in the revised paper. This update has minimal influence on the regional averages, since we have abundant data in MW region. This update yields slightly lower column averaged CO2 in HIL, which better match the CT 2015 (see updated figure below). Another update has been made on aircraft based column CO2

to fully account for the influence of the temporally uneven sampling frequency. In addition, there are other reasons that model cannot produce perfect results, for example, the transport error may also play a part in the mismatches.

[Figure]

17.      L443: I would call 4 ppm a large difference. Please remove "only" from this sentence and quantify the "large spatial gradients" observed below 2 km.

We will remove 'only'. 4 ppm is smaller compared to the 6 ppm in the SCIAMACHY retrievals in Europe. Spatial gradients below 2 km are discussed in details in section 3.1, and presented in Figure 4, with a maximum difference of ~15.5 ppm between MW and SM in summer on long-term averaged basis.

Technical comments:

We have made changes according to the reviewer suggests.

Reply to Reviewer:

Thank you for reviewing our paper.

General comments:

In general, this paper is well-written and is a useful contribution to the literature. I have only a couple serious issues with the paper. First, since this paper is about columns, it would be useful to show direct comparisons of their aircraft-derived $XCO_2$ to TCCON, to get a sense of differences in remote sensing with respect to a similarly accurate measurement. Second, sometimes they seem to argue that nearly all spatial variability in XCO2 comes from large-scale + remote fluxes and transport, rather than regional-scale fluxes. Other times (such as in the main body), they argue that a significant contribution comes from large-scale + remote fluxes and transport, but a significant contribution also comes from more regional-scale fluxes (within North America in this study). The latter is more in agreement with their data and specifically their removal experiment of section 3.4, so we suggest they rewrite the paper to stress that both mechanisms affect the horizontal structure of XCO2.

We agree with the reviewer that it will be interesting to compare TCCON with aircraft + CT based total column CO2. The following figure can give us some ideas about this comparison. TCCON values in this figure are daily averages computed from mid-day data (10 to 15 LST). Outliers in TCCON data are filtered out, by using 3.S.D threshold of residuals (with iteration) from a quadratic fit on data for each day.

[Figure]

TCCON Data DOI       10.14291/tccon.ggg2014.lamont01.R0/1149159

TCCON reference       Wunch, D., G. C. Toon, J.-F. L. Blavier, R. A. Washenfelder, J. Notholt, B. J. Connor, D. W. T. Griffith, V. Sherlock, and P. O. Wennberg (2011), The total carbon column observing network, Philosophical Transactions of the Royal Society - Series A: Mathematical, Physical and Engineering Sciences, 369(1943), 2087-2112, doi:10.1098/rsta.2010.0240. Available from: http://dx.doi.org/10.1098/rsta.2010.0240

However, it is not an apple-to-apple comparison between our aircraft-based column $CO_2$ and TCCON. The sampling time and location are different. Aircraft sampling typically takes ~ 30 min to get the whole profile through downward spiral, mostly within 0.1° of the site location. For TCCON that sampling the whole column instantly at an angle, the sampling time and the actual sampling location (especially at high altitudes), may be far away from our aircraft measurements. Mismatches caused by both should be examined carefully if we want answers for accuracy, which should be the topic of another in-depth study. In addition, we need to acknowledge that the aircraft and tower measurements are calibrated on WMO scale; however, direct calibrations of remote sensing instruments are not available. It makes more sense to compare TCCON against calibrated measurements for accuracy instead of the other way around.

Regarding to reviewer comments on large-scale + remote fluxes and transport:
We do not mean that regional/local fluxes have no influence on long-term averaged column $CO_2$ spatial pattern. But the major contributor is transport, instead of surface flux. In addition to our observation that long-term column $CO_2$ spatial gradients pattern is smooth, and mainly reflects large-scale circulation pattern, our results from CT experiments showing that 50% of North-South gradient are due to Eurasia flux, which is stated in both abstract and conclusion parts. Since Eurasia boreal region is obviously not the only upwind location with significant $CO_2$ sources, (upwind Canadian region is likely contribute to transport signal too), we actually expected **larger than 50%** transport influence in total. We clarify this point in the revised paper.

Though the authors discuss at length their use of CT2015 to extend the aircraft profiles from 330 hPa to the top-of-atmosphere, they don't really discuss the potential errors from the fact that the aircraft measurements don't go all the way to the surface. While some sites do sample very close to the surface (e.g. 0.2 km at SGP), other sites don't even sample as low as 1 km AGL (e.g. CAR: 2.2 km agl, HIL: 1.1 km agl). Because the concentration can change quickly near the surface, the authors need to acknowledge this source of error, and ideally estimate its potential magnitude. They could do this by taking full CT profiles for their sites and comparing the column extended their way (lowest value held constant) vs. the CT value. They can also estimate it based on tower observations for a few sites.

The reviewer has mistaken the unit we use for elevation and altitude. We use 'above sea level (asl)' instead of 'above ground level (agl)' in our study, and the information is presented in S Table 1. CAR is located on a high elevation terrain, as sated in Line 143. There is no < 1km (asl) measurements because the averaged surface elevation is already 1.5 km asl. The lowest layer of aircraft measurements at CAR is ~ 2.1 km asl, which is ~ 0.6 km agl. HIL does sample below 1 km asl; the lowest level s is ~ 600 m asl (surface elevation is ~200 m asl). Most regions we discuss in this study also have tower measurements below 1km asl, although aircrafts can sample even lower than the highest inlets of some towers, as we can see from Fig. 5 and Fig. 6. We have a few layers of measurements below 800 hPa for all regions (except MC with 1.5km surface elevation), as we can see from in Fig.5 and Fig.6.

Regarding to the reviewer's comment that "the concentration can change quickly near the surface", is the reviewer referring to nighttime measurements? The aircraft profiles are taken at late morning and early afternoon (data outside 10:00-17:00 local time are excluded in this study), during which time the PBL are generally well-mixed. The tower data we use in our study are also daytime data only, which are stated in our section 2.1. We have observed consistent daytime $CO_2$ from different heights in tower measurements, even during summer. The following figures are from our tower data: https://www.esrl.noaa.gov/gmd/ccgg/about/co2_measurements.html

[Figure]

*Figure 5a. Measurements of $CO_2$ by a continuous in-situ analyzer at a tall tower during the summer in northern Wisconsin. The top plot show the $CO_2$ values at six different heights on the tower.*

[Figure]

*Figure 5b. Measurements of $CO_2$ by a continuous in-situ analyzer at a tall tower during the winter in northern Wisconsin.*

Finally, their section on Reuter et al. (2014, hereafter R14) doesn't fully fit in with the rest of the paper. Specifically, they show that CarbonTracker (CT) agrees well with aircraft over North America. This is not surprising, given the excellent coverage of surface sites in North America, which CT assimilates. However, this does NOT guarantee that CT will be right everywhere, which they seem to imply throughout the paper. They need to stress that this work validates CT in North America specifically. Anywhere else, and they are merely speculating. Also, they focus on the "hot spots" in the R14 map, which are likely due to inhomogenous SCIAMACHY spatiotemporal sampling as much as anything, and may disappear in a regional inversion. Regarding the European sink, the stations that CT assimilates are all in Western Europe, and much of the controversy is really how much sink is portioned between the Europe and Eurasian Boreal Transcom regions, as discussed in both R14 and Reuter et al. (2016). Their central argument that the spatial patterns of XCO₂ from R14 are unphysical is qualitative at best, and, from this author's perspective, partially based on the unfortunate color scheme (rainbow) chosen by R14 which can accentuate very small spatial differences. They should tone down the language to something more like to say that the spatial patterns seem unphysical, but more work would be required to really rebut their physicality. They also need to cite Reuter et al. (2016), which gives a thorough overview of the "European sink controversy".

The reviewer agrees that the column CO2 spatial pattern from SCIAMACHY retrievals is "unphysical". From our comparison with CarbonTracker, which is proven to be trustworthy for the long-term spatial pattern over the U.S., we think the column CO2 spatial pattern from SCIAMACHY retrievals is unrealistic and disagree with our understanding of transport. It is important to point out these in our study.

We think it is reasonable to believe that CT produces trustworthy results even outside North America, for the following reasons: (1) CT-Europe that simulate more surface measurements shows similar results as CT. (2) The transport in CT is reasonably well, supported by the good simulation of SF6, a tracer for large-scale transport (see https://www.esrl.noaa.gov/gmd/ccgg/carbontracker/CT2016_doc.php#tth_sEc6, section 6.2). (3) The reviewer seems to suggest that we cannot judge the model permanence in Europe because we don't have the truth to compare with, unlike the well-calibrated aircraft profiles we have in North America. But that's the case for all CO2 models over Europe. At least for CT, it compared well with N.A. aircraft measurements, even though aircraft measurements are not assimilated in CT. This gives us more confidence for the CT model.

Regrading to the spatial patterns of SCIAMACHY in Reuter et al.(2014), it is unlikely the inhomogeneous spatiotemporal sampling is the major reason for spotted feature, since 8 years of data are used to generate this spatial pattern. We do not propose that the conclusion of R14 about the large European carbon sink is wrong. However, it is important to point out the serious issue that this study use unrealistic spatial pattern. Based on our knowledge of transport, a smooth spatial pattern is expected, especially after averaging 8 years of data. We do not agree with the reviewer that the "unfortunate color scheme (rainbow) chosen by R14" is the reason for the unrealistic spatial pattern. We use the same rainbow color scheme and the same 6 ppm maximum color scale as R14 in our Fig. 11, in order to compare with Reuter et al. We do not see the spotted feature on similar spatial resolution; instead we see gradual spatial patterns that match our understanding on transport (Keppel-Aleks et al., 2012 find similar results about the significant influence of large-scale transport).

We will cite Reuter et al. 2016 for European sink. However, the main purpose of our study is not to discuss the European sink. We intend to show a smooth spatial pattern of column CO2 (based on well-calibrated measurements) that match our understanding of transport, which should be a baseline for current and future remote sensing retrievals.

**Specific and technical comments:**

L54: Suggest you add language such as "or else regional-scale biases can result" and cite Chevallier et al. (2014).

We cite the Chevallier et al. (2014) in the Line 88.

L65: While satellite retrievals of $XCO_2$ can certainly have regional biases, the Inoue et al. (2016) paper is hardly thorough. It discusses one particular retrieval (the NIES retrieval) of GOSAT data, and in fact the authors appear to cherry-pick the biases of 2.09 and 3.37 ppm they list, which come from stations with the fewest colocations (3 and 1, respectively). Most of the stations in that paper have biases less than 1ppm, consistent with their comparison to TCCON, and that of Kulawik et al (2016). The authors should give a fairer representation of Inoue et al, including their TCCON comparisons, and mention the mean biases relative to TCCON given in Kulawik et al. (2016).

The reviewer suggests that comparison between GOSAT and TCCON is fairer. The Inoue et al., 2016 compared GOSAT retrievals with aircraft based column CO2 after correcting GOSAT retrievals against TCCON. This is independent evaluation; however, comparison with TCCON after correction (using TCCON data as reference) is not. We don't think comparing with TCCON is a better approach than comparing with aircraft-based column CO2. While TCCON shares some similarities with satellite retrievals column CO2 considering both use remote sensing technique, TCCON itself has known bias. TCCON cannot be directly calibrated, unlike aircraft profiles. Well-calibrated measurements should be a prefer 'truth' to compare with.

We will provide the information that 20 out of 27 stations in Inoue et al. (2016) study shows difference smaller than 1 ppm compared with aircraft-based column CO2.

L88: To the 2001 and 2004 papers they cite, they should add Chevallier et al. (2014), which more explicitly demonstrate this.

We cite the Chevallier et al. (2014) in revised paper.

L119-125: Regarding the recently-discovered high humidity bias. To me, this seems like a pretty large error, especially if the mean / typical correction is 0.5 ppm! Since this is the first paper to discuss it, a plot seems warranted – authors should plot the estimated error based on humidity, and justify this 1.7% humidity cut-off value they give. Also, the authors aren't entirely clear on the units of "humidity" here. Presumably they mean volumetric mixing ratio = dry air mole fraction; please be more clear, or use a more typical unit (such as mass mixing ratio or specific humidity).

While a 0.5 ppm correction seems to be large, only ~ 4% of total aircraft measurements or ~ 12% of those below 2 km are impacted. The 1.7 % threshold is preliminary, based on data from tower sites; however, our more recent aircraft measurements with larger amount of data find that this threshold is sufficient for data correction. We are still working to better understand and quantify this bias, and testing dryer design to avoid influence of high humidity. To evaluate the impact of this bias on our study, we have compared the partial column CO2 results (below 330 hPa) before and after the corrections. Please see figures below:

[Figure]

From this comparison, it is safe to say that this correction does not pose an obvious difference on our results. However, for future studies that are not averaging over long-term record like this study, we need to be prudent on this bias.

The unit for humidity here is ppm in mole fraction, which is the mole of water vapor versus whole air. Dry air should not be assumed when we talk about ratio of mole fractions. We clarify the unit in revised paper.

L158: I think the authors mean Figure 3, not 2.

We change it to "Fig. 3".

L277 – Here is where the authors should discuss potential errors by extending the lowest measurement to the surface (see discussion above in *General Comments*).

Please see response to general comment.

L308 – Please remind the reader here what "partial column" means. I think surface to 330 hPa, but it's not entirely clear.

We clarify "partial column" in revised paper.

L318 – Suggest you change the sentence to stress that CT captures these features of XCO2 *over North America* reasonably well. There is absolutely no guarantee that this will be true elsewhere, as discussed in *General Comments*. You can speculate that it will work in other locations, but you don't show it, so as written this statement is not justified.

We modify the sentence. Please also see response to general comment.

L329 – Please define "drawdown". Does it mean peak to trough of CO2 in the seasonal cycle, or the minimum CO2 relative to MLO? The former is more typically accepted as the definition of drawdown. Then make Fig 8 consistent with your chosen definition.

"Drawdown" refers to the variation amplitudes in the seasonal CO2 signals that mostly resulted from strong plants activity. We clarify this point several places in the revised paper.

L346 – Again emphasize that the agreement is only shown for North America.

Please see response to general comment.

L348-350 Regarding CT not showing hotspots. This is not necessarily surprising and could be due to CT's limited resolution. I've attached an unofficial plot of the CAMS nrt XCO2 over North America, for the June-August 2016 average (see e.g. Massart et al. 2016 for details), which has a resolution of roughly 0.25°, and hotspots over L.A. and Mexico city are clearly visible, consistent with the findings of Kort et al. (2012) and others.

[Figure]

It is reasonable that finer grid can show more pronounce hot spots. However, from the CAMS results for 2016 summer, we cannot see random spotted feature like the 8-year mean SCIAMACHY retrievals shows (which are in a much coarser grid).

It is interesting to see whether the hot-spot in LA and Mexico city also exist in the a priori. How does the long-term averaged spatial pattern look like from CAMS? Is the hot-spot feature still pronounced? The long-term averages are those that both Reuter et al. 2014 and our study focus on.

L374-6 : This implies that local/regional fluxes account for ~50% of the horizontal gradient in this case. Authors should mention this fact, rather than argue that virtually all of the spatial gradients come from large scale transport, which is what they state in the abstract.

Please see response to general comment.

L392: Reuter et al (2014) doesn't only use SCIAMACHY; they use GOSAT as well, and in fact, they use a number of different GOSAT retrievals, not just one, and get consistent results from all of them. Statements in this paper should reflect this.

The unrealistic spatial patterns in Reuter et al. (2014) Fig. 2 are SCIAMACHY retrievals. The main purpose of our study is not to discuss the European sink. We intend to show a smooth spatial pattern of column $CO_2$ that match our understanding of transport, which should be a baseline for remote sensing retrievals. Since our study is not an inversion study, we cannot prove that the inversion results in Reuter et al. (2014) are wrong. We do not intent to focus on the resulting fluxes. Instead, we focus on pointing out that the spatial pattern is unrealistic. The spatial pattern from GOSAT is actually drastically different from SCIAMACHY. Spatial gradient is the backbone of inversion modeling that resolve surface sources and sinks. Their consistent results from different inversions that use drastically different spatial patterns are not assuring, instead, they raise the question that why different inversions based on drastically different spatial gradients (or the unrealistic gradients in SCIAMACHY) can produce consistent results. It is not clear that whether the data are serving as good enough constraints in inversions or the a priori and transport are dominating the results.

L409: The Feng et al. (2016) study is only applicable to global inversions. R14 uses a regional inversion with only observations in Europe, to make themselves insensitive to an overall regional bias, and only use the gradients in the satellite data to infer the European sink. This is somewhat at odds with the Feng et al. study, which shows that much of the apparent European sink comes from satellite observations outside of Europe.

We do not agree with the reviewer that Feng et al. study only applicable to global inversion. From the prospect of inversion setup, there is no guarantee that regional inversion is better than global inversion since reliable boundary condition is still needed for regional inversion. Global constraints are needed to be satisfied. From the perspective of regional satellite data, using regional data makes the inversion less susceptible to extreme bias, such as land-sea bias; however, there are still potential sources of bias within the European region which may be caused by spatial differences in clouds and aerosols, surface albedo, etc. Small bias is enough to produce drastically fluxes results. Previous study by Masarie et al. (2011) had evaluated the impact of CO2 measurement bias on CarbonTracker flux estimates and found that 1 ppm bias at one site, the Park Fall ,Wisconsin (LEF site in our study), can cause 68 Tg C/yr bias in flux estimate for Temperate North America (~ 10% of the estimated North American annual terrestrial uptake). Flux estimate errors are also found in Europe and boreal Eurasia to compensate for the errors in North America. The importance of small bias is also shown in RT14 Figure 2c that an overall 0.5 ppm spatial gradient over Europe is enough to produce the extra European carbon sink.

[revised manuscript text omitted]

---

## Referee Report (RR1)

Updated review of Lan et al, 2017.

The paper is generally improved over the previous version, and will be ready for publication after the following minor revisions are made, and one major revision is considered.

This review is ordered from most significant to least.

- The abstract makes this statement: "Our results confirm that continental-scale total column XCO2 gradients simulated by CarbonTracker are realisitic and can be used to evaluate the credibility of spatial patterns from satellite retrievals.

  First, the authors only demonstrate that it is reliable in North America. No statements within the paper address why it should be reliable everywhere, considering that far more data are assimilated into CT from North America than anywhere else. The authors made such statements in the response to reviewers – these needed to be added to the paper to corroborate this claim.

  Second, the authors make no mention of potential sampling bias in the observations. In the case of their focus on the SCIAMACHY observations over Europe, section 3.5 still makes no mention of the possible sampling biases in the observations. The authors are sampling all data points over Europe all the time from their model. SCIAMACHY only makes measurements when it flies overhead, and when it is sufficiently clear (and this is true of all the CO2-measuring sensors). When you average these irregular observations together, *you do not get a mean spatial pattern.* You get a mis-mash that includes whatever samples you happened to take. The authors need to mention this in the revised manuscript as a possible explanation of the appearance of an unphysical spatial pattern in the Reuter et al (2014) figure. How the data are used/assimilated is the critical factor. If the data are used ignoring this fact, it is of course a problem. But nearly all inversion systems sample the data at the times and places of the observations, so this effect is at least partially taken into account. Therefore, the claim in the abstract that CT can be used to evaluate mean spatial patterns from satellites is dubious, since *they are simply not the same thing.* The only way to get around this is issue to sample the model like the satellite, which the authors currently do not do nor discuss. For the paper to be acceptable, this statement in the abstract, and all related statements throughout the paper, needs to be either eliminated or qualified with this caveat.

- Regarding the comparison to TCCON, this comparison should at least be in the supplementary materials, along with error statistics. People have been comparing TCCON to aircraft for a long time and while it is not perfectly apples-to-apples, it can give a good idea of consistency. The authors saying "we don't need to do this" is unacceptable to this reviewer, considering they've already done it, they just need to include it.

- Regarding the error analysis and the missing of the surface layers, it was a misunderstanding on my part, so I withdraw that criticism.

- Line 434: Again the authors need to stress that CT is realistic over North America. Please rewrite to be: "Since spatial gradients from *CT2015 have been shown to be realistic on continental scales over North America*, boreal fluxes inferred by CT2015 ... *may* be more trustworthy than...."

- Line 436: "However, the European carbon sink is still *elusive;*".

---

## Author Response (AR2)

**Response to reviewers**

First, we would like to thank the reviewers for their valuable time, efforts, and advice.

Response to reviewer #2:

*1. "My main concern is that two of the three original reviewers requested comparisons with TCCON, and the authors have done so, but have not included this in the revised manuscript nor in the supplementary materials."*
*"I agree that what you have plotted is not an apples-to-apples comparison. I would suggest that instead, you average the TCCON data during the 30 minute aircraft sampling times instead of simply taking mid-day averages. I also do not see the need for your additional 3-sigma filtering."*

We have provided the TCCON data comparisons in the supporting information section of the revised manuscript, which also include the averaged TCCON data within 30 min. of aircraft measurements. Please see below:

[Figure]

**Fig. S2** Comparisons between aircraft and CT based total column $CO_2$ with TCCON total column $CO_2$. TCCON data with Solar Zenith Angle larger than $60^0$ are not included in these comparisons. (a), (b), and (c) are time series, residuals (TCCON - aircraft_CT_extended), and scatter plot comparisons between SGP and TCCON Lamont site ($36.60^0$N, $-97.49^0$E), respectively. (d), (e), and (f) are the same between LEF and TCCON Park Falls site ($45.94^0$N, $-90.27^0$E). In (a) and (d), TCCON midday values are the averages between 10:00 to 15:00 LST; TCCON 30 min. values are the averages within 30 min. of aircraft sampling time. TCCON 30 min. values are used in (b), (c), (e), and (f).

*2. We stated in the previous response that "In addition, the aircraft and tower measurements are calibrated on WMO scale; however, direct calibrations of remote sensing instruments are not available. It makes more sense to compare TCCON against calibrated measurements for accuracy instead of the other way around. Our study focuses on the long-term mean spatial patterns of column CO2 over North America and the influence of large-scale transport..."*

*Reviewer: "The overall scaling of the two data products is not of prime interest, here. As you mention, the long-term mean spatial pattern are primarily important, which I believe you can assess using a comparison with TCCON data. This might give you a way of assessing the CT stratosphere, and improving your product!"*

We are not talking about overall (global) scaling, but those that depend on regional and seasonal conditions. We have evaluated the CT stratosphere simulations using calibrated measurements of AirCore, which shows reasonable agreements. However, TCCON data cannot be calibrated directly. That's why we opt to use AirCore for comparison.

*3. We stated in the previous response that "For TCCON that sampling the whole column instantly at an angle, the actual sampling location, especially at high altitudes, may be far away from our aircraft inlets even though the distance on the ground is much smaller."*

*Reviewer: Near noon, this should be a less significant problem. At the very least, this is a quantifiable problem, and you have not demonstrated the problem in a quantitative manner. Also, you base much of your paper on the fact that the stratosphere does not change significantly in time or space, so this argument is not strong. (For example, you later state: "There is little spatial variation of atmospheric CO2 in the stratosphere over mid-latitude region (Andrews et al., 2001)." and "We cannot properly assign an exact number as the uncertainty for the top 1/3 of the atmosphere without routine in-situ measurements. However, spatial gradient and atmospheric variability is very small in this part of the atmosphere, and thus it is not important in our study.")*

The TCCON midday values are also provided in the revised manuscript. Our study focus on spatial gradients of the total column; as we can see from the Fig. 4d, Fig. 6, and Fig. 7, the spatial differences in the stratospheric CO2 are very small compared with the tropospheric CO2 (especially within the PBL). Thus, it is expected that the stratospheric CO2 does not contribute significantly to the spatial differences of the total column.

Response to reviewer #3:

- The abstract makes this statement: "Our results confirm that continental-scale total column XCO2 gradients simulated by CarbonTracker are realisitic and can be used to evaluate the credibility of spatial patterns from satellite retrievals.

  First, the authors only demonstrate that it is reliable in North America. No statements within the paper address why it should be reliable everywhere, considering that far more data are assimilated into CT from North America than anywhere else. The authors made such statements in the response to reviewers – these needed to be added to the paper to corroborate this claim.

- Line 434: Again the authors need to stress that CT is realistic over North America. Please rewrite to be: "Since spatial gradients from *CT2015 have been shown to be realistic on continental scales over North America*, boreal fluxes inferred by CT2015 ... *may* be more trustworthy than...."

We have already answered this comment in the previous response. We admit that our study focuses on North America; however, our vigorous comparison between calibrated data and CT results has shown the significant influence of transport on column $CO_2$ gradients, and found smooth column $CO_2$ gradients. These conclusions should also apply in Europe. We don't think the transport over Western Europe can be so different from that over North America that it could explain the larger gradient and the hot spots in Reuter et al. (2014) Fig. 2. Our response was already provided (and copied here); the reviewer hasn't commented on our response:

"We think it is reasonable to believe that CT produces trustworthy results even outside North America, for the following reasons: (1) CT-Europe that simulate more surface measurements shows similar results as CT. (2) The transport in CT is reasonably well, supported by the good simulation of $SF_6$, a tracer for large-scale transport (see https://www.esrl.noaa.gov/gmd/ccgg/carbontracker/CT2016_doc.php#tth_sEc6, section 6.2). (3) The reviewer seems to suggest that we cannot judge the model performance in Europe because we don't have the truth to compare with, unlike the well-calibrated aircraft profiles we have in North America. But that's the case for all $CO_2$ models over Europe. At least for CT, it compared well with N.A. aircraft measurements, even though aircraft measurements are not assimilated in CT. This gives us more confidence for the CT model."

In addition, CarbonTracker also shows reasonable agreement with TCCON sites in Europe (e.g. https://www.esrl.noaa.gov/gmd/ccgg/carbontracker/tccon.php?site=bremen01), and its performance is comparable with (or even better than some) other models (e.g. https://www.esrl.noaa.gov/gmd/ccgg/carbontracker/OCO2_insitu_rev1/tccon.php?ds=bremen&r unid=IS).

Second, the authors make no mention of potential sampling bias in the observations. In the case of their focus on the SCIAMACHY observations over Europe, section 3.5 still makes no mention of the possible sampling biases in the observations. The authors are sampling all data points over Europe all the time from their model. SCIAMACHY only makes measurements when it flies overhead, and when it is sufficiently clear (and this is true of all the CO2-measuring sensors). When you average these irregular observations together, *you do not get a mean spatial pattern.* You get a mis-mash that includes whatever samples you happened to take. The authors need to mention this in the revised manuscript as a possible explanation of the appearance of an unphysical spatial pattern in the Reuter et al (2014) figure. How the data are used/assimilated is the critical factor. If the data are used ignoring this fact, it is of course a problem. But nearly all inversion systems sample the data at the times and places of the observations, so this effect is at least partially taken into account. Therefore, the claim in the abstract that CT can be used to evaluate mean spatial patterns from satellites is dubious, since *they are simply not the same thing.* The only way to get around this is issue to sample the model like the satellite, which the authors currently do not do nor discuss. For the paper to be acceptable, this statement in the abstract, and all related statements throughout the paper, needs to be either eliminated or qualified with this caveat.

The reviewer suggests that SCIAMACHY BESD (2003-2010 June-August) pattern in Reuter et al., 2014 is not a mean pattern (then Reuter et al., 2014 Fig. 2 is misleading to show this pattern as an 8-year statistics) and the sampling bias could be responsible for the unphysical column $CO_2$ pattern. We show here the data coverage of SCIAMACHY for 3 summers (2005/2006/2007), and the 3-year averaged column $CO_2$ pattern from sampling the CT2015 using SCIAMACHY latitude/longitude/date (see Figure below). SCIAMACHY retrievals provide good data coverage for most regions south of $55^0$N in Europe (except that United Kingdom and Ireland have less coverage). The averaged column $CO_2$ from CT2015 shows smaller gradients and smoother pattern, compared with SCIAMACHY BESD (Reuter et al., 2014 Fig. 2); this 3-year pattern already agrees with our understand of large-scale transport and meridional $CO_2$ gradients. Thus the sampling bias is unlikely the main reason for the large spatial difference of up to 6 ppm between Belgium, Netherland, north of Germany and south Ukraine and Kazakhstan in their figure. We are actually not sure on the specific causes for the unphysical pattern, thus we didn't commenting on this in our manuscript. But we will provide the above information regarding to the sampling bias in the revised manuscript and add the figure (d) in supporting material section. What we are sure about is that SCIAMACHY does not provide a credible spatial $CO_2$ pattern and we think it should be pointed out. The reviewer seems to be confirming that, in fact, SCIAMACHY's sampling biases prevent scientists from being able to infer reliable sources/sinks. We agree with that.

[Figure]

Figure for reviewing process. SCIAMACHY BESD data coverages for 2005 (**a**), 2006 (**b**), and 2007 (**c**) summer (JJA), and averaged (over three summers) column $CO_2$ pattern from sampling CT2015 with SCIAMACHY latitude/longitude/date for these periods (**d**). Black circles show the locations of extreme hot/cold spots with up to 6 ppm differences from Reuter et al. (2014), Fig.2. For (**d**), seasonal mean is first removed for each year before combining 3-year data together and averaged within a $2^0 \times 2^0$ grid. Girds with less than 10 soundings for three summers are excluded; color scale shows maximum 6 ppm difference, similar as Reuter et al. (2014), Fig.2. SCIAMACHY latitude/longitude/date information is attained from http://www.esa-ghg-cci.org/sites/default/files/documents/public/documents/GHG-CCI_DATA.html. SCIAMACHY BESD data coverages for other years are accessible at: http://www.esa-ghg-cci.org/?q=webfm_send/200.

Then, the reviewer states that how the data are assimilated is a critical factor. If 8-year data still cannot provide realistic spatial gradients, then the inversion needs to be based on a priori assumptions and model transport, while a large portion of satellite measured radiances have to be ignored. We have already made this point in our previous response; however, the reviewer didn't comment on it. In the meantime one loses track of the evidence: (1) The influence of prior assumptions has to be tuned relative to the retrievals derived from the radiances. (2)The assimilation process entrains transport biases of the transport models into the results. It is well established that all of these models have their own biases. That is why we are comparing the performance of CarbonTracker to calibrated aircraft and tower data, not to other transport models, nor to TCCON. Unlike the reviewer's statement about sampling the model like satellite, the "only way to get around the issue" of biases is to also compare other models and TCCON to calibrated data. They are the only data that stand on their own feet, no models needed.

- Regarding the comparison to TCCON, this comparison should at least be in the supplementary materials, along with error statistics. People have been comparing TCCON to aircraft for a long time and while it is not perfectly apples-to-apples, it can give a good idea of consistency. The authors saying "we don't need to do this" is unacceptable to this reviewer, considering they've already done it, they just need to include it.

We will provide the TCCON comparisons with aircraft + Carbon Tracker based column $CO_2$ in the supporting material:

[Figure]

**Fig. S2** Comparisons between aircraft and CT based total column $CO_2$ with TCCON total column $CO_2$. TCCON data with Solar Zenith Angle larger than $60^0$ are not included in these comparisons. (a), (b), and (c) are time series, residuals (TCCON - aircraft_CT_extended), and scatter plot comparisons between SGP and TCCON Lamont site ($36.60^0$N, $-97.49^0$E), respectively. (d), (e), and (f) are the same between LEF and TCCON Park Falls site (45.94$^0$N, -90.27$^0$E). In (a) and (d), TCCON midday values are the averages between 10:00 to 15:00 LST; TCCON 30 min. values are the averages within 30 min. of aircraft sampling time. TCCON 30 min. values are used in (b), (c), (e), and (f).

- Line 436: "However, the European carbon sink is still *elusive;*".

We will make the change in the revised paper.

[revised manuscript text omitted]

---

## Author Response (AR3)

**Response to editor**

Dear editor,

    Thank you for editing our manuscript. Regarding to your comment:

*1) the sampling of CT2015 CO2 fields does not take into account any clear sky criterion (only then can columns be retrieved from the satellite) except for matching the date and location. I do see still potential for a larger sampling bias. In the model the cloud cover could well be different. Also if the sampling was not made during the same hour of the day (rather than just during the same date), a bias can arise. This should be clarified and included in section 3.5 of the manuscript.*

    We actually sampled the CT2015 within 1 hour of the SCIAMACHY BESD data (version 02.00.08, same as in Reuter et al., 2014). We matched the latitude/longitude/hour when there's available SCIAMCHY BESD data (the columns were already retrieved, so model cloud is not a concern in this case). We will clarify this point in our revised manuscript.

*2) The patterns in the CT2005 columns when sampling at SCIAMACHY locations/dates shown in the "Figure for reviewing process" are also not smooth, in fact in terms of smoothness they are not so different from those shown in Fig. 2a in Reuter et al., 2015, and they are more different from the smooth patterns in Fig. 11. This indicates a rather strong impact of a sampling bias, even when no additional criterion for cloudiness or hour of day is used.*

    The patterns we provided in the "Figure for reviewing process" is a 3-year average, unlike the 8-year average in Reuter et al. (2014) or the 11 year average in the Fig.11. Thus the color looks less smooth overall. However, our focuses are the gradients and the locations of strong hot/cold spots. The 3-year average already shows much smaller spatial difference (in ppm) compared with Reuter et al. (2014), and we cannot find cold/hot spots in those locations as in Reuter et al., Fig 2. To make our case stronger, we have updated the 3-year average to an 8-year average map in our revised manuscript. In this case our sampling period and data points can perfectly match those in Reuter et al., 2014, Fig. 2. The 8-year average column $CO_2$ shows a maximum 3-4 ppm difference and a gradual pattern, similar as our Fig.11 (see figure below). We understand that sampling bias has played a role in the Reuter et al. (2014) Figure 2, especially over Ireland, UK, and south Sweden regions with very little amount of samples (it is surprising that these areas are included in Reuter et al., 2014 Fig. 2); however, it cannot explain the large differences and the strong hot/cold spots over the circled areas (hot spots over Belgium, Netherland, north of Germany, and low spots over the Ukraine and Kazakhstan). In addition, when we compare CT2015 at the time and place of our measured aircraft profiles with the measured data over the North America (lower sampling frequency for most aircraft sites compared with satellite), we do not see such strong biases (our Fig. 7 and Fig. 9).

[Figure]

(Reuter et al., 2014, Fig. 2)

2003-2010 Jun-Aug pattern from sampling CT2015 with SCIAMACHY BESD latitude/longitude/hour. Jun-Aug mean for each year is first subtracted before putting 8-year data together and averaged within each grid (2 ⁰ × 2⁰). Grids with less than 24 soundings during 8 summers are excluded.

Our Fig. 11, the Mauna Loa deseasonalized trend is subtracted in this figure, but the color scale is set to reflect maximum 6 ppm difference.

*Later on you state in your reply: "The reviewer seems to be confirming that, in fact, SCIAMACHY's sampling biases prevent scientists from being able to infer reliable sources/sinks." This is completely in contrast to how I interpret this reviewers comment. All the reviewer is saying is that so far you haven't been able to prove that the satellite retrievals are unphysical, as sampling biases can potentially explain this. I share this opinion, and I would think unless a sampling bias can be excluded as a reason, the language in section 3.5 needs to be toned down significantly.*

We will provide the above 2003-2010 averages figure in supporting information in the revised manuscript. This figure clearly demonstrates that the sampling bias cannot explain the 6 ppm difference over Belgium, Netherland, north of Germany, versus Ukraine and Kazakhstan.

[revised manuscript text omitted]

---

## Author Response (AR4)

**Response to editor**

Dear editor,

Thank you for editing our manuscript.

Regarding to your comment:
*The evidence is that CT2015 uses ground-based observations with good coverage over North America, but with limited coverage over Europe, to estimate posterior fluxes. Forward modeling of CO2 based on these posterior fluxes compares favorable with aircraft profile observations over North America, but not over Europe where such profile data are not available (as also pointed out by the reviewers). Any sampling bias resulting from the sparse surface network coverage over Europe was not taken into account; such a bias was shown by Reuter et al. (2014) to lead to significantly smaller net uptake estimates over Europe.*

Our rigorous comparison between calibrated data (including ground and aircraft measurements) and CT2015 has shown that CT2015 is trustworthy over North America, and that's why we have now also compared CT2015 with SCIAMACHY BESD over North America. The following figures show that the differences between CT2015 and SCIAMACHY BESD are up to ~ 3 ppm over North America. Differences of this magnitude imply large differences in inferred $CO_2$ sources and sinks. We are deeply skeptical about the SCIAMACHY BESD data. We also find up to 3 ppm differences over Europe. We will provide the following figure in Supplement section, and corresponding statements in the text.

We note that, although the dense data coverage obtainable from satellite soundings is often touted as a great advantage, after the necessary data selection for the optimal conditions that minimize (but not eliminate) biases the remaining coverage is remarkably spotty and noisy in adjacent grid cells even for an eight-year three-month summer average.

[Figure]

**Figure S8.** Differences between SCIAMACHY BESD column $CO_2$ and CT2015 over North America (left panel) and Europe (right Panel) for 2003-2010 June-August averages. CT2015 is sampled using only accepted SCIAMACHY BESD soundings (latitude/longitude/hour). Residuals (SCIAMACHY BESD – CT2015) are averaged within a $2^0 \times 2^0$ grid. Grid cells with less than 24 soundings for 8 summers are excluded.

SCIAMACHY data is attained from:
http://www.esa-ghg-cci.org/sites/default/files/documents/public/documents/GHG-CCI_DATA.html.
SCIAMACHY BESD reference:
Reuter, M., Bovensmann, H., Buchwitz, M., Burrows, J. P., Connor, B. J., Deutscher, N. M., Griffith, D. W. T., Heymann, J., Keppel-Aleks, G., Messerschmidt, J., Notholt, J., Petri, C., Robinson, J., Schneising, O., Sherlock, V., Velazco, V., Warneke, T., Wennberg, P. O. and Wunch, D.: Retrieval of atmospheric CO2 with enhanced accuracy and precision from SCIAMACHY: Validation with FTS measurements and comparison with model results, J. Geophys. Res. Atmos., 116, 2011.

*When sensing sampling bias of remote sensing observations is taken into account by sampling the model at the correct locations and times, the pattern of CT2015 columns change, but this does not explain the difference to patterns in SCIAMACHY derived columns over Europe. Actual filtering using only clear sky periods based on cloud cover information from the meteorological fields used to drive CT2015 was not investigated nor was any potential effect on the CT2015 cerived patterns in CO2 columns mentioned.*

The cloud cover should not influence the comparison between SCIAMACHY BESD and the CT2015 pattern from sampling CT2015 using available SCIAMACHY BESD lat/lon/hour. The SCIAMAHCY BESD has already taken account of the cloud cover issue. Regarding to the differences between a full CT2015 pattern and the CT2015 with SICAMACHY BESD lat/lon/hour, we could evaluate the cloud cover impacts on the differences by looking into the cloud cover information from CT2015 transport model. However, this approach may not give us useful information regarding the SCIAMACHY BESD biases themselves, since the cloud cover in the CT transport model is different from the actual cloud conditions that SCIAMACHY BESD uses. More importantly, it is not the focus of our study to dig into the reasons why the SCIAMACHY BESD patterns are unrealistic. The SCIAMACHY BESD data provider should explain the unphysical column $CO_2$ pattern and why their data (with significant sampling bias and other biases) can be used in modelling to yield credible results.

*The strong wording in section 3.5, specifically lines 434 – 436, is not justified given the evidence. If this is not modified to better reflect the limited evidence, I have no choice but to reject the paper.*

We have removed the lines 434-436 in the revised version.

[revised manuscript text omitted]